# Dissipation of electron-beam-driven plasma wakes

Rafal Zgadzaj[1], T. Silva[2], V. K. Khudyakov[3,4], A. Sosedkin[3,4], J. Allen[5], S. Gessner[5], Zhengyan Li[1,6], M. Litos[5,7], J. Vieira[2], K. V. Lotov[3,4], M. J. Hogan[5], V. Yakimenko[5] & M. C. Downer[1✉]

Metre-scale plasma wakefield accelerators have imparted energy gain approaching 10 giga-electronvolts to single nano-Coulomb electron bunches. To reach useful average currents, however, the enormous energy density that the driver deposits into the wake must be removed efficiently between shots. Yet mechanisms by which wakes dissipate their energy into surrounding plasma remain poorly understood. Here, we report picosecond-time-resolved, grazing-angle optical shadowgraphic measurements and large-scale particle-in-cell simulations of ion channels emerging from broken wakes that electron bunches from the SLAC linac generate in tenuous lithium plasma. Measurements show the channel boundary expands radially at 1 million metres-per-second for over a nanosecond. Simulations show that ions and electrons that the original wake propels outward, carrying 90 percent of its energy, drive this expansion by impact-ionizing surrounding neutral lithium. The results provide a basis for understanding global thermodynamics of multi-GeV plasma accelerators, which underlie their viability for applications demanding high average beam current.

[1] University of Texas at Austin, 1 University Station C1600, Austin, TX 78712-1081, USA. [2] GoLP/Instituto de Plasmas e Fusão Nuclear-Laboratório Associado, Insituto Superior Técnico, Lisboa, Portugal. [3] Budker Institute of Nuclear Physics, 630090 Novosibirsk, Russia. [4] Novosibirsk State University, 630090 Novosibirsk, Russia. [5] SLAC National Accelerator Laboratory, Menlo Park, CA 94025, USA. [6]Present address: Huazhong University of Science and Technology, Wuhan, China. [7]Present address: Center for Integrated Plasma Studies, University of Colorado Boulder, Boulder, CO 80309, USA. ✉email: downer@physics.utexas.edu

Localized deposition of concentrated energy into plasmas by particle bunches or laser pulses underlies fast ignition of laser fusion[1], formation of plasma waveguides[2], and wakefield acceleration of electrons and positrons[3]. Subrelativistic particles or laser pulses deposit energy into dense uniform plasmas primarily via collisions[2], analogous to ohmically heating a resistor, but collisions become inefficient for tenuous plasma and/or relativistic excitation. Relativistic particle bunches (or laser pulses), on the other hand, excite tenuous plasma by creating electron density structures, such as channels[4] or Langmuir waves[3], via Coulomb (or ponderomotive) forces, analogous to charging a capacitor. Previous experiments in plasmas of millimeter (mm) length and atmospheric electron density ($n_e \sim 10^{19}$ cm$^{-3}$) have documented the initial (first few ps) stage of transferring energy stored initially in such structures into long-term ion motion[4,5].

The emergence of quasi-monoenergetic multi-GeV plasma-wakefield accelerators, in which relativistic particle bunches[6,7] (or laser pulses[8]) deposit energy into strongly nonlinear wakes in plasma of density $n_e \sim 10^{17}$ cm$^{-3}$, however, raises the question of energy dissipation with renewed urgency. Multi-GeV plasma accelerators require such low $n_e$ in order to avoid depleting the driver's energy before it propagates ~1 m, the length of plasma wake needed to accelerate electrons or positrons to multi-GeV energy[3]. Heat removal by flowing the medium supersonically transversely to the driver propagation direction, which is routine for mm-long, 10 μm wide, high $n_e$, gas-jet-based MeV plasma accelerators[3,5], is not feasible for meter long, 100 μm wide, low $n_e$, multi-GeV accelerators, which require stationary vessels to confine their long, tenuous plasmas[6–8]. Such accelerators will therefore require new strategies for managing deposited heat, based on quantitative understanding of energy dissipation. Such understanding is fundamental to achieving luminosities large enough to observe rare processes at the energy frontier. Accelerators achieve high luminosity by delivering focused high-charge, high-energy beams at high repetition rate. Planned conventional machines such as the International Linear Collider[9] or the Compact Linear Collider[10], for example, are designed to deliver ~10 megawatt (MW) to the interaction point. Current conceptual designs for plasma-based accelerators aim to achieve comparable luminosity by accelerating ~nC bunches at ~10 kHz repetition rate (i.e. ~100 μs interbunch spacing)[11,12]. This need has spurred worldwide efforts to develop petawatt-peak-power drive lasers that can operate at multi-kHz repetition rates[13]. However, the problem of dissipating, or otherwise managing, deposited power of order tens of kW per meter over ~100 m of plasma has received comparatively little attention. The present study aims to understand the fundamental processes by which plasma accelerator structures at $n_e \sim 10^{17}$ cm$^{-3}$ dissipate their energy into surrounding plasma, and to evaluate their global energy budget over a nanosecond (ns) time scale. It thus builds a thermodynamic foundation on which future engineering solutions of the heat management problem can be based.

To produce high-quality bunches, plasma accelerators of any $n_e$ usually operate in a strongly nonlinear regime in which the driver blows out a "bubble"-like cavity devoid of plasma electrons in its immediate wake[3]. In such structures the energy density $|E|^2/(2\epsilon_0)$ of internal wake fields $E$ approaches the rest energy density $n_e mc^2$ of plasma electrons[14]. Simulations predict that such electron wakes can spawn ion wakes of unique structure and dynamics[14–16], early stages of which were observed[5] at high $n_e$. However, no experiments at any $n_e$ have yet explored how the enormous energy density stored in a nonlinear wake redistributes over nanoseconds among accelerated electrons, undirected hot electrons, freely streaming ions, radiation, electrostatic fields, and ionization of surrounding gas, as well as collective ion motion. Understanding this complex process at $n_e \sim 10^{17}$ cm$^{-3}$ demands experiments with

precisely characterized multi-GeV drive bunches (or petawatt laser pulses), probes that track particle and energy flow over millimeters, and simulation of multifarious plasma processes over nanoseconds. The scale and complexity of the problem rival those of other energy transport problems involving tenuous plasma, such as heating of the solar corona[17] and acceleration of energetic cosmic rays[18].

Here we present ps-time-resolved optical shadowgraphic measurements of meter-length plasma columns that emerge from broken, highly nonlinear plasma wakes that energetic electron bunches generate in self-ionized, initially neutral lithium vapor. As a drive bunch propagates ~0.3 m into the vapor, its center and trailing edge self-focus to high density, enabling them to field-ionize a wide plasma column and to drive a strongly nonlinear plasma wake well within its boundaries. The drive bunch then reaches steady state, and continues generating this nonlinear wake over the next meter. A diagnostic optical pulse probes the expanding plasma column at a fixed longitudinal location $z$ within this steady-state region at time delays $0 < \Delta t < 1.5$ ns in 0.1 ns intervals, and coarsely out to 10 μs. These observations serves as a calorimeter that determines the fraction of the initial wake energy that the plasma column retains after the wake breaks. Simulations reveal that the initial wake transfers energy into the surrounding medium via the following sequence of events: the initial wake breaks, expelling fast electrons from the plasma; radial electric fields arise that propel ions outward at tens of keV while escorting electrons; outwardly streaming electrons and ions ionize and excite surrounding neutral lithium, expanding plasma volume several hundred-fold. Benchmarking simulated plasma expansion against measurements quantifies energy retention in the plasma column and elucidates mechanisms that drive its expansion.

## Results

**Generation of nonlinear wakes.** Experiments were carried out at the SLAC Facility for Advanced aCcelerator Experimental Tests (FACET)[19]. The first 2 km of the SLAC linac delivered drive electron bunches (e-bunches) of energy 20 GeV, charge 2.0 nC, rms radius $\sigma_r = 30$ μm, length $\sigma_z = 55$ μm to the entrance of the interaction region. Unlike small-scale plasma wakefield accelerator experiments, in which parameters of drive e-bunches delivered from a tabletop laser wakefield accelerator could only be estimated[5], here e-bunches from SLAC were characterized with high precision (see "Methods"). This is important for simulating subsequent plasma dynamics accurately over an ns time scale. Drive bunches entered a 150-cm-long column of Li vapor, in which a 120-cm-long region of uniform atomic density $n_a = 8 \times 10^{16}$ cm$^{-3}$ was centered between 15-cm-long entrance and exit density ramps. This vapor was generated and contained within a cylindrical heat-pipe oven[6,20] of 1.6 cm radius (see "Methods"), and provided the medium for plasma formation and wake excitation.

Particle-in-cell (PIC) simulations discussed below showed that the SLAC e-bunches singly field-ionized the Li vapor over its entire length out to initial radius $r(0) \approx 40$ μm and electron density $n_e = n_a$, and drove a strongly nonlinear electron density wake consisting of a train of nearly fully blown-out cavities of radius $\lambdabar_p = 20$ μm, propagating at ~$c$ along the axis of the resulting plasma. Thus the initial plasma column was wide enough to fully support the generated wake. Under these conditions, this wake closely resembles the corresponding wake that a similar e-bunch would form in a pre-ionized plasma of similar density[21]. The main differences are that, in the self-ionized case, the wake forms further back in the drive bunch profile[21] and decays more rapidly along $z$ due to head

erosion[22,23]. For our conditions, ~5% of the drive bunch erodes over 1 m of propagation (see Supplementary Eq. (1)). Neither difference has any bearing on fixed $z$, ns-scale, transverse plasma expansion dynamics of interest here. These dynamics are determined by the amplitude, structure, and stored energy of the initial wake at the longitudinal location $z$ at which we measure and simulate them, regardless of whether that wake formed in self- or finite-radius pre-ionized plasma.

The wake cavities enclose accelerating fields of magnitude $E \sim m\omega_{\mathrm{p}}c/e \sim 1$ GV/cm, where $\omega_{\mathrm{p}} = [n_e e^2/(\epsilon_0 m)]^{1/2}$ is the electron plasma frequency. Although we have the capability at FACET to pre-ionize the lithium vapor along the entire drive beam path out to $r \sim 500\,\mu$m[24], here we retained the Li vapor blanket as an in situ medium for recording energy transport out of the directly excited wake. As discussed below, outwardly streaming ions and electrons, which carry away most of the wake's energy, ionize neutral Li atoms, inducing a large change in the vapor's refractive index $\eta$. This enables us to detect the ionization front with ~15 ps time and <40 μm space resolution (both limited primarily by imaging resolution) using a probe pulse. Ionization of the Li blanket consumes <1% of outwardly streaming particle energy during the first ~1 ns, and thus does not perturb overall energy transport significantly. Consequently the wake dissipation dynamics measured and simulated here in self-ionized plasma also reflect those of an equivalent wake generated in pre-ionized plasma of similar finite radius.

When SLAC delivered 2 nC, 20 GeV (i.e. 40 J total energy) drive bunches to the interaction region at repetition rate 1 Hz or less, no heat-pipe temperature rise, nor other time-dependent behavior, was observed upon turning on the beam. Downstream measurements of spent driver energy (see Supplementary Fig. 2) show that ~60% of electrons in each bunch contribute to forming a plasma wake, each losing 11% of its energy in so doing—i.e. each bunch deposits 7% of its total incident energy, or 2.6 J, into the 1.2 m-long plasma wake (2.2 J/m). Thus at 1 Hz, the plasma acquires energy at 2.6 W, which is only 0.2% of typical oven heater power (1300 W). When, on the other hand, we increased repetition rate to ~10 Hz, oven temperature typically rose tens of degrees within minutes, even though this repetition rate is ~1000× lower than current design projections[11]. Here we report data acquired at 1 Hz to avoid drifts in oven temperature during extended data acquisition runs. No witness e-bunch was injected with the driver. Moreover, non-intercepting beam charge monitors positioned before and after the plasma detected no change in beam charge (see "Methods"), suggesting that the wake trapped negligible charge from background plasma. Indeed, in view of the near-light speed of the driver and the sub-$10^{17}$ cm$^{-3}$ plasma density, trapping and acceleration of background electrons requires an external trigger, such as an optical injection pulse, under these conditions[25]. Thus plasma wave energy was dissipated entirely within the medium.

**Measurement of plasma expansion**. To diagnose the evolving radial profile of the plasma column, a collimated probe laser pulse (wavelength $\lambda = 0.8\,\mu$m, duration $\tau = 0.1$ ps, transverse width 0.5 cm FWHM) that was electronically synchronized with the drive e-bunch with ~0.1 ps jitter entered one end of the heat pipe 0.8 cm off-center. It then propagated through a 100-cm-long central section of the Li column at angle $\theta = 8$ mrad to the e-bunch propagation direction [see Fig. 1a, b] at time delay $-1$ ps $< \Delta t < 10\,\mu$s after the e-bunch, before exiting the other end on the opposite side of center. By using this grazing $\theta$, the largest that the long narrow heat-pipe oven allowed without clipping the probe pulse, we probed the tenuous plasma profile ~100× more sensitively than with a transverse ($\theta = \pi/2$ rad) probe, because of the

long interaction length, at the mild cost of averaging longitudinal ($z$) density variations $n_e(z)$ over $\Delta z \sim 0.5$ cm. Moreover, as discussed below, with this geometry we maximized sensitivity to the region of interest—the expanding profile's advancing outer edge—at the cost of insensitivity to its internal structures. Use of an e-bunch driver eliminated strong depolarized, forward-directed supercontinuum that a laser-driven nonlinear wake generates[26], which is extremely challenging to discriminate from probe light in this near co-propagating pump-probe geometry.

A lens imaged the portion of the transmitted probe pulse that it collected within its $f/40$ cone from a vacuum object plane at longitudinal position $z = 75$ cm near the center of the heat pipe (labeled O in Fig. 1a) to a charge-coupled device (CCD) camera. Here $z = 0$ refers to the foot of the entrance density ramp. Images had ~1 cm depth of field and unity magnification. Unavoidable obstructions blocked ~1/4 of the circular lens aperture [see gray area, Fig. 1b], which influenced some nonessential details of images, as discussed below. When the probe arrived before the e-bunch ($\Delta t < 0$), only the incident probe pulse profile was observed. At $\Delta t > 0$, an approximately parabola-shaped shadow outlined with alternating bright and dark fringes, resulting from probe refraction and diffraction from the e-bunch-excited plasma column, was observed. Figure 1b shows schematically how the shadow/fringe pattern evolves for fixed $\Delta t$ as the probe propagates through the near field of the plasma column. Figure 1c shows how it evolves in the far field as $\Delta t$ varies from 100 to 1200 ps. The parabolic shadow expands at ~$10^6$ m/s, eventually exceeding the camera's field of view for $\Delta t > 1200$ ps. Figure 1d–f shows calculated images for three simulations of plasma evolution, discussed below and in "Methods". One horizontal position in these patterns (highlighted by vertical white dashed line in Fig. 1c–f) corresponds to plane O. Fringed regions straddling this plane are out-of-focus images of upstream ($z < 75$ cm) and downstream ($z > 75$ cm) slices of the column. They represent images of near-field diffraction patterns of this obliquely illuminated column at each $z$[27]. At small $\Delta t$, the vertex of the parabola appears to be located at $z = 75$ cm [see 100 ps images in Fig. 1c–f], i.e. the position in the images where the object plane intersects the plasma column. In fact, the vertex is shifted slightly to $z = 75$ cm $+ \Delta z$. As the plasma column widens, $\Delta z$ increases [see 100 to 1200 ps images in Fig. 1c, f]. This is the result of increasing refraction in the cylindrical plasma column.

Figure 1g shows an image at the longest accessible delay $\Delta t = 10\,\mu$s. The uniformly dark image shows that the plasma column continues to refract probe light out of the imaging lens collection cone long after the column expands beyond the field of view. Here, we focus on the first 1.3 ns of plasma expansion, a range accessible to PIC simulations.

Figure 2a–d illustrates via probe propagation simulations (see "Methods") how details of the images in Fig. 1c–f arise. Figure 2a shows a half cross-section of the plasma column's refractive index profile $\eta(r)$ at a representative $\Delta t$, with contours (black circles) in $\Delta \eta = 10^{-5}$ increments superposed. Figure 2b shows the corresponding intensity half-profile of the probe pulse at the same $z$-plane. In this picture, the probe emerges along the page normal, while the plasma column axis is tilted 8 mrad from left (behind page) to right (in front of page). A parabolic shadow (left) develops as probe light refracts away from columns axis, after penetrating to the $\eta - 1 = 10^{-5}$ contour in the plane of incidence. Fringes (right) approximately parallel to the shadow boundary develop as refracted probe light interferes with un-deflected incident probe light. This probe profile is, however, not directly observed. Rather, a lens relays it to a CCD. For an ideal unobstructed lens and cylindrically symmetric plasma column, the detector would record the profile shown in Fig. 2c. Because the probe passes through the remainder of the plasma column,

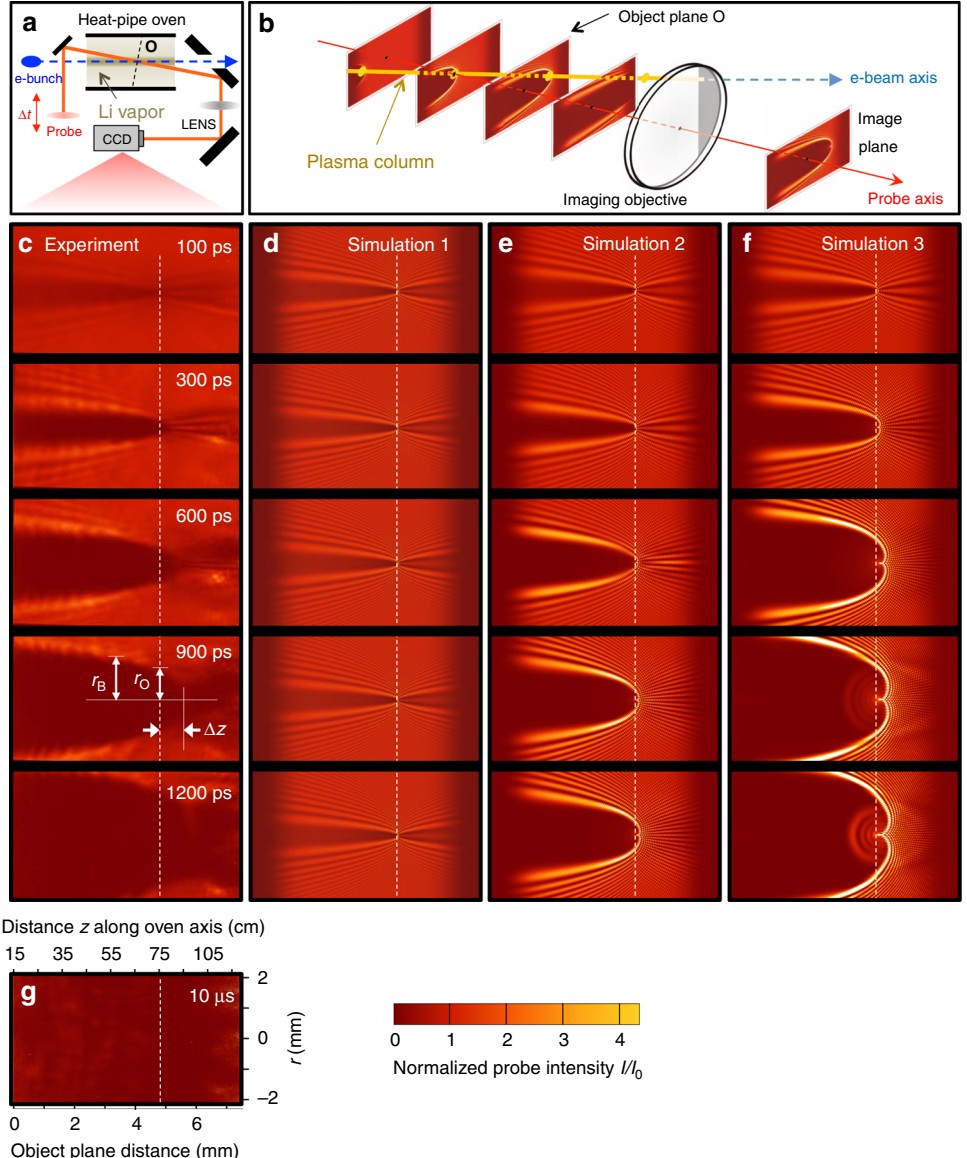

**Fig. 1 Imaging of expanding Li$^+$ ion column following electron wake excitation. a** Overview of experimental setup, showing path of laser probe pulse (orange solid line) through plasma column (dashed blue line) at grazing angle 8 mrad with variable time delay $\Delta t$ after e-bunch excitation. The resulting diffraction pattern is imaged from vacuum object plane (O) onto a CCD. **b** Schematic depiction of evolving probe intensity profile as it passes obliquely through plasma column and imaging lens. An obstruction blocked the gray area on the lens. **c** Experimental probe images for $\Delta t = 100, 300, 600, 900,$ and 1200 ps, normalized to unperturbed probe intensity $I_0$ (see color scale, lower right), averaged over 30 shots. See Supplementary Fig. 1 for comparative single-shot images. Dashed vertical white lines: intersection of plane O with plasma column. Each probe image is 4 mm high ×7.5 mm wide; horizontal dimension corresponds to projected distance 1.0 m along plasma column axis. **d–f** Simulated probe images for three plasma expansion models: **d** including dynamics of ions within initial plasma column only; **e** including impact ionization of ground state neutral lithium surrounding initial plasma column; **f** including impact excitation of neutral lithium to 2P, 3S, and 3P states and impact ionization. **g** Experimental probe image at $\Delta t = 10$ μs, for which plasma column expanded well beyond field of view. Horizontal and vertical scales in **g** also apply to each image in panels **c–f**.

the image is distorted in two ways from the probe's in situ shape. First, a nephroid-shaped cusp singularity forms in the image of the parabola's vertex, similar to bright optical caustics that form within the shadow of an obliquely illuminated drinking glass[28]. Second, a second set of interference fringes approximately orthogonal to the shadow boundary develops outside the shadow.

The first of these features proved sensitive to the above-mentioned partial blockage of the lens aperture. Figure 2d shows the reshaped shadow vertex that results when probe propagation calculations take this blockage into account. The second of these features proved sensitive to slight deviations of the simulated column from cylindrical symmetry, and to small longitudinal

non-uniformities. Thus these two features were seldom observed in actual images (see Fig. 1c). Nevertheless, the vertex shift $\Delta z$, along with shadow radii $r_O$ at O and $r_B$ at another selected longitudinal location B, as marked in Fig. 1c, were consistently observed. Orange data points and gray uncertainty ranges in Fig. 2e–g show evolving values $r_O(\Delta t)$, $r_B(\Delta t)$, and $\Delta z(\Delta t)$, respectively, for $0 < \Delta t < 1200$ ps, where B here corresponds to $z = 55$ cm. Black and orange curves in Fig. 2h–k show single-shot and 30-shot averaged lineouts, respectively, of shadows at O (Fig. 2h, j) and B (Fig. 2i, k) for $\Delta t = 100$ ps (Fig. 2h, i) and 1200 ps (Fig. 2j, k). Fringes outside of, and parallel to, the shadow boundary were observed frequently in both single-shot (Fig. 2i,

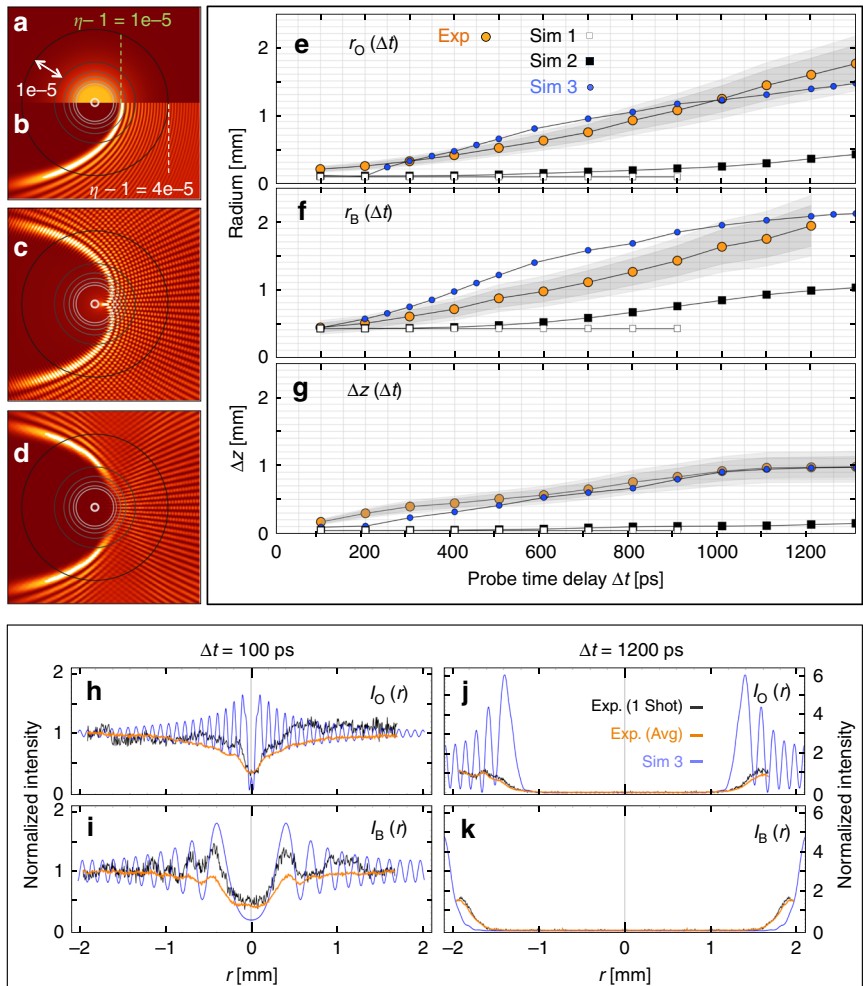

**Fig. 2 Comparison of probe image features with simulation results. a–d** Electromagnetic simulations showing how probe intensity profile evolves through plasma column to detector: **a** Typical refractive index cross-section $\eta(r)$ of plasma column at object plane O, with column axis tilted at 8 mrad to page normal from left (back) to right (front); black circles: index contours in $\Delta\eta = 10^{-5}$ increments; **b** radial intensity profile of probe, propagating normally out of page, at O, showing light penetration to $\eta - 1 = 10^{-5}$ contour, parabolic shadow and surrounding interference fringes; **c** corresponding probe profile after ideal imaging to detector, showing axial caustic and orthogonal interference fringes acquired in passing through remainder of plasma column. **d** Corresponding probe profile after non-ideal imaging to detector, showing re-shaping of vertex region due to partial blockage of imaging lens shown in Fig. 1b. **e–g** Plots of **e** $r_O(\Delta t)$, **f** $r_B(\Delta t)$, and **g** $\Delta z(\Delta t)$ from measured probe images (orange-filled circle data points), showing $1\sigma$ (dark gray) and $2\sigma$ (light gray band) variations over 30 shots, and from the three theoretical models corresponding to Fig. 1d–f: no impact ionization (open gray squares), ground state impact ionization (filled black squares), impact excitation+impact ionization (filled blue circles). **h–k** Lineouts of measured single shot (black curves) and 30-shot averaged (orange curves) normalized probe intensity at positions O (**h**, **j**) and B (**i**, **k**) for $\Delta t = 100$ ps (**h**, **i**) and 1200 ps (**j**, **k**), compared to Simulation 3 calculations (blue dashed curves).

black curve) and multi-shot averaged (Fig. 1c) data. Probe propagation simulations showed that fringe spacing was sensitive to $\theta$, but agreed well with observed fringe spacing for $\theta = 8.0 \pm 0.1$ mrad, thus corroborating the independently measured probe angle. However, observed fringe contrast is invariably lower than calculated, due to imperfect symmetry of the plasma column. In addition, fringes sometimes wash out upon multi-shot averaging, because of shot-to-shot fluctuations in $\theta$ and in drive bunch intensity. Thus, $r_O(\Delta t)$, $r_B(\Delta t)$, and $\Delta z(\Delta t)$ constituted the most robust observables for quantitative comparison with simulations.

**Qualitative plasma expansion mechanisms.** Results in Fig. 2e, f show that empirical radii $r_O(\Delta t)$ and $r_B(\Delta t)$ grow on average at $1.4 \times 10^6$ m/s over 1.3 ns, and even accelerate slightly during this interval. Probe propagation simulations show that these radii are approximately twice the plasma column radius $r_p$, here defined as the radius at which $n_e(r_p) \approx 0.2 n_a$, implying that $r_p$ expands at

near-constant velocity $v_p \approx 0.7 \times 10^6$ m/s. Such kinetics rule out expansion driven by electron heat or radiative transport[29], since heat front velocity would decrease rapidly with time[4]. They also cannot be explained by a radial shock wave at the ion acoustic velocity, which is initially only ~$10^4$ m/s for our conditions, and would also decrease within ~1 ns as electron temperature cools[2]. Rather, the observed near-constant $v_p$ must be attributed to an ionization front driven by charged particles, particularly high-momentum Li ions, streaming freely into surrounding Li vapor[4]. Li ions propagating at the observed $v_p$, for example, would have energy $E_i \sim 20$ keV. These could be produced if average outward radial electrostatic fields of order $0.01 < \langle E_r \rangle < 0.05$ GV/cm acted on the ions over radial distance of order $200 > r > 40$ µm (or time interval $50 > \tau > 10$ ps) in the collapsed electron wake following its excitation. Such ions have mean free paths of several mm in Li vapor of $n_a = 0.8 \times 10^{17}$ cm$^{-3}$, experience only small angle elastic scattering, and lose energy to neutral atoms primarily via impact excitation and ionization, which entails loss per impact only up to

the first ionization energy (~5.4 eV) of Li. These characteristics are consistent with near-constant-velocity expansion over a ns time scale. In addition, the ions escort electrons, which maintain charge quasi-neutrality and assist in exciting and ionizing Li atoms.

**Simulations of plasma expansion.** To understand plasma expansion quantitatively, we carried out PIC simulations of Li plasma dynamics out to $\Delta t \sim 1.3$ ns using two complementary PIC codes OSIRIS and LCODE. We modeled ionization, self-focusing of the drive bunch, electron wake excitation, and early ($\Delta t \lesssim 40$ ps) electron wake and ion dynamics in a fully self-consistent manner using OSIRIS[30] in cylindrical geometry (see "Methods" for details). These simulations modeled radial acceleration of ions and electrons by electrostatic fields of the collapsing electron wake with high space and time resolution, before these particles began to interact significantly with surrounding Li gas. OSIRIS simulations tracked self-consistent driver and plasma evolution for 15 cm of propagation through the gas density up-ramp and an additional 16.9 cm into the 120-cm-long density plateau (i.e. up to $z = 31.9$ cm). No particle trapping was observed at $z \lesssim 31.9$ cm up to $\Delta t = 40$ ps of wake evolution. Trapping can thus be assumed negligible at larger $\Delta t$, when wake amplitude and electron energy are lower.

To simulate long-term plasma evolution, we input the compressed e-bunch and plasma profiles from the OSIRIS simulation output at $z = 31.9$ cm into the quasistatic, axisymmetric LCODE[31] as initial conditions. LCODE then simulated plasma dynamics at this fixed $z$, which do not depend on continuing evolution of the drive bunch in the downstream plasma (i.e. at $z > 31.9$ cm). Consequently, the bunch evolution no longer needed to be tracked, enabling time-efficient simulation of long-term plasma dynamics out to $\Delta t \approx 1.3$ ns. Moreover, impact ionization of neutral gas by outwardly streaming electrons and ions comes into play on this time scale, and in fact dominates plasma expansion for $\Delta t > 100$ ps. Accordingly, we introduced well-established collisional/ionization schemes into LCODE that have been extensively implemented and tested in other codes[32,33]. Specifically, we included the most important elementary collisional processes—elastic binary collisions[32,34] and impact ionization of neutral lithium atoms by electrons[35] and fast lithium ions[36]. For electron-impact ionization, we included single-step ionization from the neutral lithium ground state and two-step ionization[37,38] via $2P$, $3S$, and $3P$ excited states. Conditions simulated here do not involve highly relativistic collisions, high density, or other extreme parameters, and thus fall well within previously tested precision limits of these collision/ionization schemes. Nevertheless, because of the special importance of collisional fast electron deceleration in determining the expansion rate of the plasma column, we carried out a new test to ensure that the collision scheme implemented in LCODE reproduced the deceleration rate of 1–4 keV electrons predicted by analytical theory[39] with high accuracy (see "Methods").

Figure 3 shows OSIRIS simulation results. The electron bunch, initially of bi-Gaussian $r$, $z$ profile (see Fig. 3a), ionized Li according to the Ammosov–Delone–Krainov (ADK) model[40]. Figure 3b shows the driver (blue) and plasma (orange) after the former propagated to $z = 31.9$ cm. The bunch's leading edge was not dense enough to ionize Li, and thus propagated as in vacuum. Ionization started in the denser center of the bunch, which then drove a nonlinear plasma wake, which in turn compressed the beam waist radially. In Fig. 3b, the bunch's trailing edge has compressed to ~100 times its initial density. This enabled it to singly-ionize Li gas completely out to $r \approx 40$ μm, and partially out

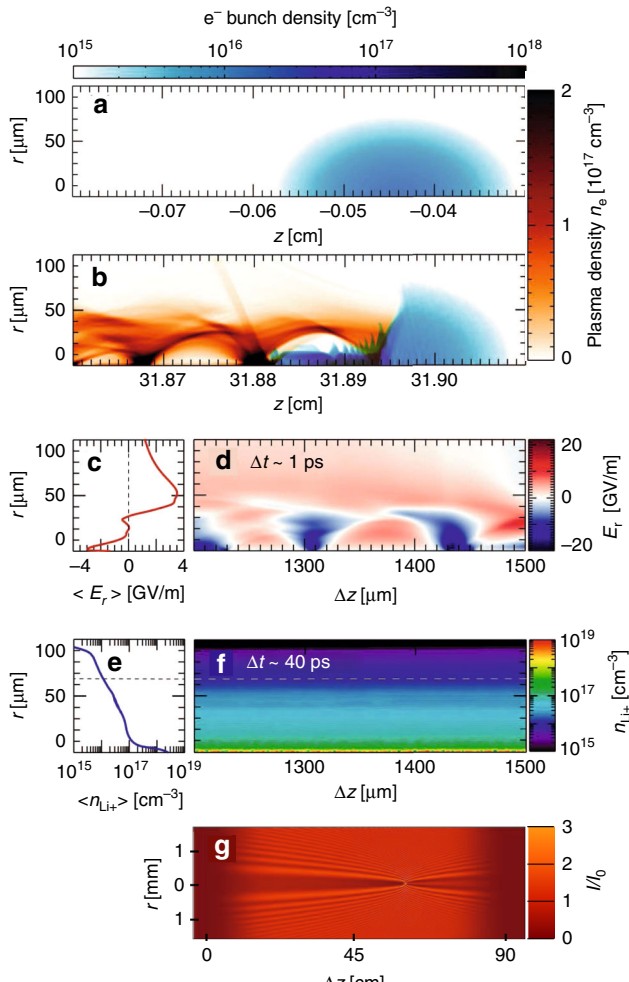

**Fig. 3 OSIRIS simulation results.** Drive bunch profile at **a** $z < 0$ (blue), before entering Li vapor at $z = 0$, and at **b** $z \approx 32$ cm, after entering the Li density plateau, focusing, and creating trailing beam-ionized plasma and electron wake (orange). **c** Fixed-window simulation of longitudinally averaged and **d** full radial electric field $E_r(r, z)$ profiles at $\Delta t \sim 1$ ps after the driver passes. The non-zero averaged field [red curve in **c**] attracts (repels) ions at $r \lesssim 30$ μm ($r > 30$ μm) to (from) the axis. **e** Longitudinally-averaged and **f** full ion density at $\Delta t \sim 40$ ps after driver passage. Dashed line indicates the maximum $r \approx 65$ μm to which drive bunch directly ionized Li. **g** Calculated probe diffraction pattern from density profile in **e**, cf. measured pattern for $\Delta t \sim 100$ ps in Fig. 1c.

to $r \approx 65$ μm, and to drive fully blown-out plasma bubbles of radius $\lambda_p = 20$ μm.

Figure 3d shows the radial electric field $E_r(r, z)$ remaining at $\Delta t = 1$ ps. Typical of the aftermath of a nonlinear wake, this field is still non-zero[15]. Because of their large mass, plasma ions initially respond to $\langle E_r(r) \rangle$, i.e. $E_r(r, z)$ averaged longitudinally over ~$6\lambda_p$, shown in Fig. 3c. Since $\langle E_r(r) \rangle$ switches sign at $r \approx 30$ μm, it attracts ions initially (i.e. before wave-breaking occurs) at $r < 30$ μm toward the axis, while pushing ions initially at $r > 30$ μm outward. The driver also expels a fraction of plasma electrons into surrounding neutral gas, leaving net positive charge in the plasma that further propels outward ion motion. The resulting ion density structure $n_{Li^+}(r, z)$, seen in Fig. 3e, f at $\Delta t = 40$ ps before (Fig. 3f) and after (Fig. 3e) longitudinal averaging, has a peak on axis. In addition, the outermost ions diffuse outward, moving from ~65 μm (dashed lines in Fig. 3e, f) to ~100 μm. Figure 3g shows the calculated probe diffraction pattern from a

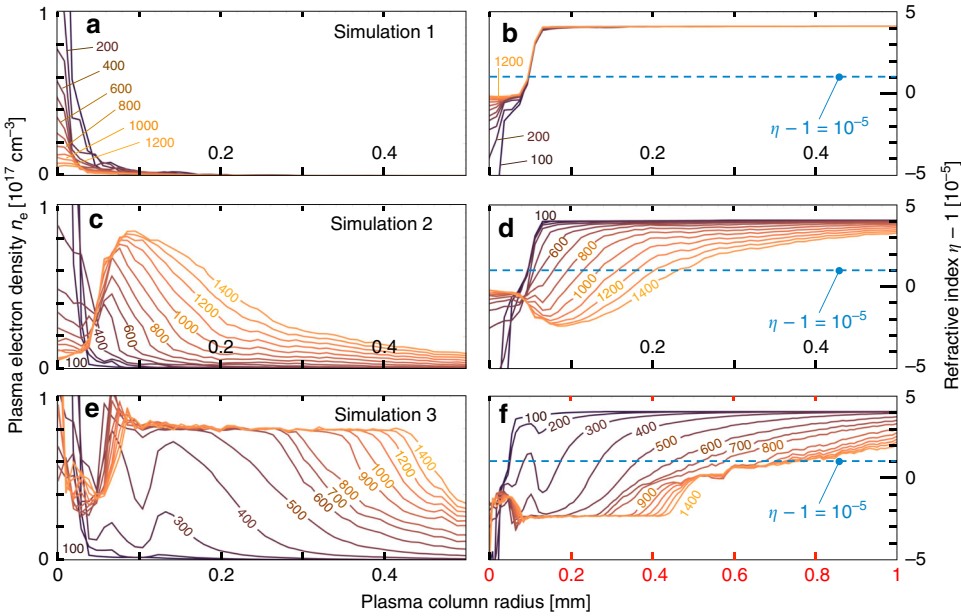

**Fig. 4 LCODE simulations of evolving plasma density and refractive index.** Comparison of evolution of radial plasma density distributions (left) and corresponding refractive index distributions (right) for the three models used in LCODE simulations. **a–b** No secondary ionization. **c–d** Single-step electron and ion-impact ionization from ground state. **e–f** Single-step plus two-step electron-impact ionization via 2P, 3S, and 3P excited states. Labels on individual curves denote $\Delta t$ in ps. Although the left-hand column plots only plasma density $n_e(r)$, neutral lithium atom (and, for Simulation 3, excited state) distributions are also non-uniform and evolving, and were taken into account in calculating refractive index distributions shown in the right-hand column. Horizontal blue dashed lines in the latter indicate the threshold index $\eta = 1.00001$ at which the probe reflects from the plasma column. Panel **f** has twice the horizontal axis range as other panels.

longitudinally uniform plasma column of the shape of Fig. 3e for our experimental geometry. It is very close to $\Delta t = 100$ ps data (Fig. 1c) and simulation (Fig. 1d). Indeed, our simulations predict negligible change in this pattern during the interval $40 < \Delta t < 100$ ps, since radially accelerated ions and electrons have not yet begun to impact-ionize surrounding Li neutrals substantially.

Figure 4 presents LCODE simulation results for $\Delta t > 100$ ps. Figure 4a shows how the plasma density profile $n_e(r, \Delta t)$ evolves over the interval $100 < \Delta t < 1200$ ps in the absence of impact ionization processes (Simulation 1). At $\Delta t = 100$ ps, the dominant feature is a sharp axial electron density maximum $n_e(r < 0.01 \text{ mm})$. This is the electron counterpart of the ion density maximum $n_{\text{Li}^+}(r < 0.01 \text{ mm})$ seen in Fig. 3e–f at $\Delta t = 40$ ps, and maintains its quasi-neutrality. Over the ensuing 1100 ps, this axial peak drops in amplitude and broadens, driven by electrostatic forces. Nevertheless, $n_e$ ($r > 40 \mu$m) never exceeds $10^{16}$ cm$^{-3}$. Figure 4b shows corresponding refractive index profiles $\eta(r, \Delta t)$ (see "Methods"). The index profile does not change noticeably for $r > 40 \mu$m throughout the simulated interval. Since the probe pulse turning radius ($\eta - 1 = 10^{-5}$, blue dashed line in Fig. 4b) occurs at $r \approx 50 \mu$m in our geometry, the probe does not sense index changes occurring at $r < 40 \mu$m (shown in Fig. 4b). Hence, Simulation 1 predicts no change in probe signatures over the interval $100 < \Delta t < 1200$ ps, as shown in Fig. 1d. Quantitative signatures $r_O(\Delta t)$, $r_B(\Delta t)$, and $\Delta z(\Delta t)$ remain unchanged over the simulation interval (see open squares in Fig. 2e, f and g, respectively). Simulation 1 encompasses ion motion physics investigated over a tens-of-ps interval in ref. [5], but does not explain longer-term expansion evident here (Figs. 1c and 2e–g).

To capture this continuing long-term expansion, we included impact ionization, induced by energetic electrons and ions streaming radially outward from the directly e-beam-ionized plasma (Fig. 3b). Figure 4c shows evolving $n_e(r, \Delta t)$ profiles when these processes are restricted to single-step ionization from the neutral Li ground state (Simulation 2). The earliest

$n_e(r, \Delta t = 100 \text{ ps})$ profile shown, and its corresponding $\eta(r, \Delta t = 100 \text{ ps})$ profile in Fig. 4d, differ only slightly from their counterparts in Simulation 1. By $\Delta t = 400$ ps, however, impact ionization has begun to create substantial new plasma in the region $50 < r < 150 \mu$m, which reaches Li vapor density (i.e. $n_e = n_a = 8 \times 10^{16}$ cm$^{-3}$) by $\Delta t = 1200$ ps (see Fig. 4c). At the microscopic level, the simulation shows that energetic electrons, which exit the original plasma first, ionize some neutral Li in this region directly, but inefficiently, since their density and ionization cross-section are small. A moving front of fast ions creates most of the new plasma. Once ions appear at a given location, more electrons come, including lower-energy plasma electrons with high impact-ionization cross-sections. Growth of this electron population triggers near-exponential plasma density growth.

The corresponding $\eta(r, \Delta t)$ for Simulation 2 also change substantially at $50 < r < 150 \mu$m (see Fig. 4d). Probe turning radius quadruples from $r \approx 50 \mu$m at $\Delta t = 400$ ps to $r \approx 200 \mu$m at $\Delta t = 1300$ ps. Consequently, Simulation 2 predicts widening probe shadows (see bottom 4 panels of Fig. 1d) with growing $r_O(\Delta t)$ and $r_B(\Delta t)$ (filled black squares, Fig. 2e, f), bringing Simulation 2 closer to the data than Simulation 1. Nevertheless, simulated $r_O(\Delta t)$ and $r_B(\Delta t)$ still fall well short of observed values (Fig. 2e, f). Moreover, Simulation 2 does not reproduce the observed shift $\Delta z(\Delta t)$ in the vertex of the parabolic shadow (see Fig. 2g).

To capture these remaining features we included two-step electron-impact ionization into LCODE (Simulation 3). In these processes, an initial electron impact excited the 2S electron of a ground state Li atom to a 2P, 3S, or 3P state. A subsequent impact within the natural lifetime of these states then ionized the atom. Including such processes increased new plasma production significantly, as $n_e(r, \Delta t)$ plots in Fig. 4e show. Simultaneously they hastened growth of probe turning radius, as $\eta(r, \Delta t)$ plots in Fig. 4f show. The partial ion density distributions $N_i(r)$ in Fig. 5a (shades of brown) show that e-impact of excited Li* atoms is, in fact, the dominant source of Li$^+$ ions at $\Delta t = 1200$ ps. Figure 5b

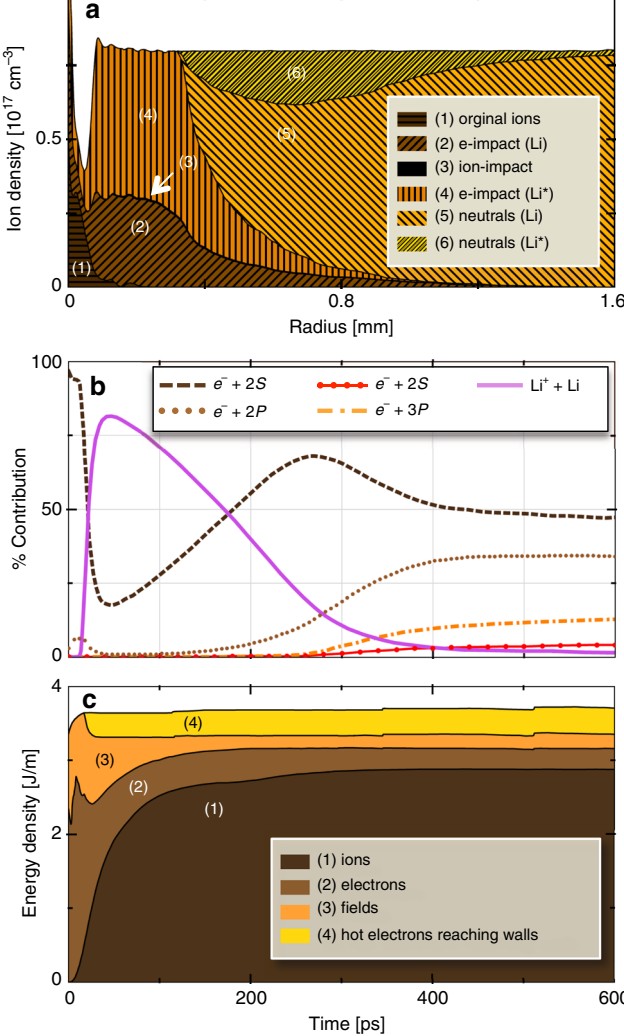

**Fig. 5 LCODE simulations of ionization and energy transport channels. a** Plot of Li$^+$ ion [(1)–(4)] and Li neutral atom [(5) and (6)] density distributions corresponding to electron density distribution at $\Delta t = 1200$ ps in Fig. 4e. Regions (1), (2), and (4) indicate relative contributions of original ions (region 1) and of electron-impact-ionized ground state (2) and excited (4) neutrals, while the barely visible region 3 (black) indicates ion-impact-ionized Li atoms. **b** Time evolution of indicated impact ionization channels. Ion-impact ionization (solid purple curve) dominates for $50 \lesssim \Delta t \lesssim 160$ ps; electron-impact ionization (dark-brown dashed, light-brown dotted, orange dot-dashed, red filled-circle curves) dominates for $\Delta t \gtrsim 160$ ps. **c** Time evolution of indicated energy transport channels. Hot electrons carry ~10% of the energy deposited in the original wake to the walls in the first ~20 ps (region 4). The expanding plasma column retains the rest without noticeable attenuation throughout the remainder of the simulated period. Electrons (2) and fields (3) carry most of the latter energy initially ($20 \lesssim \Delta t \lesssim 40$ ps), but transfer ~85% of it to radial ion motion (1) within 300 ps. The small jumps evident in curves (2)–(4) are nonphysical. They result from occasionally doubling macro-particle size and halving density as ionization increases particle number, in order to speed up the simulation.

shows, however, that two-step processes become dominant only at $\Delta t \gtrsim 400$ ps (see brown dotted, orange dot-dashed, red filled circle curves). Simulated probe images (Fig. 1f) widen more than twice as rapidly as for Simulation 2 (Fig. 1e). Moreover, a substantial vertex shift $\Delta z(1200\,\text{ps}) \approx 500\,\mu$m develops by the end of Simulation 3. Simulated average growth of $r_O(\Delta t)$ and $r_B(\Delta t)$ agrees with observed average growth over the interval

$100 < \Delta t < 1300$ ps (see Fig. 2e, f). Finally, simulated and observed probe image lineouts near the beginning (Fig. 2h, i) and end (Fig. 2j, k) of Simulation 3 agree well in width and depth, despite discrepancies in fringe amplitude. Thus Simulation 3 captures all qualitative features of the data, including the vertex shift $\Delta z(\Delta t)$ that Simulation 2 missed, as well as some key quantitative features.

## Discussion

Nevertheless, some quantitative discrepancies remain. Early in Simulation 3 ($100 < \Delta t < 600$ ps), $r_B(\Delta t)$ grows faster ($2 \times 10^6$ m/s) than observed ($1.2 \times 10^6$ m/s), resulting in $r_B$ values at $\Delta t \sim 600$ ps nearly 50% larger than observed. Later ($600 < \Delta t < 1200$ ps), on the other hand, $r_B(\Delta t)$ grows more slowly ($1.2 \times 10^6$ m/s) than observed ($1.7 \times 10^6$ m/s), yielding $r_B$ values at $\Delta t \sim 1200$ ps that agree well with observations. Thus, radial expansion of simulated (observed) images decelerates (accelerates) during the simulated (observed) $\Delta t$ interval.

There are several plausible reasons for these discrepancies. First, although incident drive bunches were thoroughly characterized, properties of the bunch after its trailing part focused inside the plasma (Fig. 3b) govern plasma expansion dynamics. Because plasma lensing is nonlinear, small errors in incident bunch properties can lead to large errors in focused bunch properties. For example, simulations assumed axisymmetric drive bunches, whereas ~10% asymmetries between $\sigma_x$ and $\sigma_y$, and 10-fold differences between focusing functions $\beta_x$ and $\beta_y$, were typically present at the plasma entrance, and could have led to asymmetric downstream focusing and plasma expansion. A second probe in an orthogonal plane would help to diagnose such expansion asymmetries, if present. Similarly, deviations in the longitudinal bunch shape from Gaussian, which were not well characterized, sensitively influence the intra-bunch position at which ionization and self-focusing begin, and in turn the fraction of incident bunch charge that drives a nonlinear wake. This can also lead to significant discrepancies between observed and simulated expansion rates.

In the later part of Simulation 3 ($600 < \Delta t < 1200$ ps), the radial slope $\partial\eta/\partial r$ of advancing refractive index profiles at the turning point radius shrinks rapidly (see Fig. 4f). As a result, simulated radii $r_B(\Delta t \gtrsim 900\,\text{ps})$ in Fig. 2f become sensitive to small perturbations in $\eta(r)$ profiles, and by extension to small deviations in the radial profile of the focused drive bunch. Departures of the incident drive bunch radial profile from its assumed Gaussian shape prior to plasma focusing, and depletion or re-shaping of focused drive bunches for $z > 30$ cm, which are neglected in quasistatic LCODE simulations, are possible sources of such deviations. In addition, neglected drive bunch evolution within the ~100-cm probed region imprints left–right asymmetry onto probe images beyond that currently simulated.

These residual discrepancies indicate that detailed quantitative comparison of simulated and measured ns-scale plasma dynamics will require selected improvements to both experiment and simulation, as noted above. Nevertheless, the broad agreement obtained in the spatial and temporal scale of post-wakefield expansion validates the basic plasma/atomic physics on which Simulation 3 is based. Its output can thus elucidate additional internal properties of the expanding plasma beyond those that were directly observed.

An example is the plasma's energy budget. According to Simulation 3, the fully focused drive bunch (Fig. 3b) deposits energy into the plasma at rate ~3.5 J/m (see Fig. 5c), in reasonable agreement with the average deposition rate (2.2 J/m) inferred from analysis of the spent drive bunch's energy spectrum (Supplementary Fig. 2 and Supplementary Methods). The latter

rate is expected to be lower than the former, since energy deposition peaks where the bunch fully focuses at $z = 31.9$ cm, and decreases thereafter as the drive bunch weakens, evolution that is not calculated here. Figure 5c shows additionally the first 0.6 ns of how deposited energy partitions and evolves over 1.3 ns, based on Simulation 3 results. Initially ($\Delta t \leq 1$ ps), beyond the horizontal resolution of Fig. 5, most of the deposited energy is stored in electromagnetic fields of the electron wake, with a minor fraction stored in kinetic energy of coherently moving plasma electrons at the bubble boundary, as is typical for wide bubbles[41]. After the wave breaks ($\Delta t \sim 1$ ps, Fig. 3c), about 10% of deposited energy transforms to kinetic energy of fast electrons (see Fig. 5c, yellow) that diverge radially, some forming a tail wave[42] visible in Fig. 3b. As the fastest electrons escape from the plasma, a radial charge separation field appears (see Figs. 3c and 5c, orange) that holds most remaining electrons within the plasma column[14]. In the first tens of ps (i.e. the range of OSIRIS simulations and the work in ref. [5]), electron kinetic energy (light brown) and electric field energy (orange) comprise most of the plasma column's energy. During the interval $30 < \Delta t < 300$ ps, the charge separation field accelerates ions, which acquire most of the energy by $\Delta t \sim 300$ ps (Fig. 5c, dark brown). The fastest ions accelerate beyond 400 keV (i.e. $v = 3.6 \times 10^6$ m/s). Overall 90% of the initially deposited energy remains in or near the plasma column. Similar confinement is expected for wakes in finite-radius pre-ionized columns. This contrasts with radially unbounded plasmas, in which fast electrons escape the heated region freely, cooling it to sub-keV temperatures within a few hundred wake periods[16]. Unbounded plasma, however, is impractical for colliders because of the enormous energy cost of producing it over hundreds of stages. Here, plasma energy and its partitioning among plasma species reach steady state for $\Delta t > 300$ ps, and change negligibly through the end of the Simulation 3 run at $\Delta t = 1300$ ps. This validates the experiment's premise that the Li blanket records energy transport via ionization without noticeably depleting the energy of the ionizing radiation. It also indicates that the $\sim 10^6$ m/s expansion continues unabated well beyond $\Delta t \approx 1.3$ ns.

In summary, results of this study have identified the principal physical mechanisms, and quantified the dominant dynamical pathways, by which highly nonlinear e-beam-driven wakes in finite-radius $n_e \sim 10^{17}$ cm$^{-3}$ plasmas release their stored electrostatic energy into the surrounding medium. Time-resolved optical diffractometry measurements of the expanding plasma column, in particular, prompted recognition of the critical, previously unrecognized role of ion-mediated impact ionization in driving the plasma radius outward during the first nanosecond. The results also make clear that the plasma columns internal electrostatic fields, even after the original wake breaks, remain not only the principal propellant of outward ion motion, but the principal force responsible for retaining 90% of the wake's initially deposited energy within the plasma column for over a nanosecond. The framework hereby established and validated provides a basis for modeling the global thermodynamics of multi-GeV plasma-wakefield accelerators, and for evaluating limits on their repetition rate. Relevant extensions of the experiments and simulations include investigations of laser-driven wakefield accelerators, and the use of varied probe geometries (e.g. larger $\theta$) that will enable space- and time-resolved observation of the plasma column's evolving internal structure.

## Methods

**Electron drive bunch characterization**. The first 2 km of the SLAC linac delivered e-bunches to the FACET interaction region in sector 20. A series of eight-turn toroidal current transformers along the FACET beamline measured absolute charge of each bunch with 2% accuracy. A pair of stripline beam position monitors, configured to measure bunch charge[43], positioned before and after the plasma,

measured *change* in bunch charge with sub-% accuracy. No change was detected. A synchrotron-X-ray-based spectrometer measured the energy spectrum of each incident bunch non-invasively with ~0.1% resolution. An integrated transition radiation monitor and a transverse deflecting cavity, both located just upstream of the FACET interaction region, measured transverse ($\sigma_{x,y}$) and longitudinal ($\sigma_z$) dimensions, respectively, of bunches entering the plasma with ~10 μm resolution. We measured $\sigma_{x,y}$ for every shot, $\sigma_z$ for selected shots. By modeling the beam focusing optics, beam dimensions inside the interaction region were determined with similar accuracy. Beams incident on FACET typically had asymmetric emittance $\epsilon_x \approx 10\epsilon_y$, so in order to have round beams with $\sigma_r \approx \sigma_x \approx \sigma_y \approx 30$ μm at the entrance of the plasma, we focused the beam into the plasma with beta-functions $\beta_y \approx 10\beta_x$. See ref. [44] for a detailed overview of FACET beam diagnostics.

**Lithium source**. Lithium was chosen for the accelerator medium because its low first ionization potential (5.4 eV) allows the drive bunch to singly field-ionize it easily over a 1.5 m path. A heat-pipe oven—consisting of a stainless steel cylindrical tube (length 2 m, inner radius 1.6 cm) heated along its center, and cooled at both ends—generated and confined the lithium gas. The vapor pressure of a melted lithium ingot loaded onto a stainless steel mesh ("wick") lining the inner wall of the hot center generated gas of temperature-controlled density $n_a$. Helium buffer gas concentrated at the cold ends confined it longitudinally. The longitudinal density profile $n_a(z)$ was deduced from the temperature profile $T(z)$ along the length of the oven, measured by inserting a thermocouple probe into the heat pipe[6,20]. Thermocouple scans with our normal operating heating power of 1340 W yielded a 1.2-m-long central plateau of density $n_a = 8.0 \pm 0.2 \times 10^{16}$ cm$^{-3}$, with 0.15 m long density ramps at each end (see Supplementary Fig. 1). With the heat pipe in steady-state operating mode, we controlled overall Li density primarily by adjusting buffer gas pressure, and the length of the central density plateau by adjusting heater power.

**Probe laser and imaging system**. Probe pulses (~1 mJ energy, polarized in plane of incidence) were split from the 500 mJ, 50 fs output pulse train of a 10-TW Ti:S laser system[24]. Transverse probe intensity profiles contained hot spots, which were superposed on single-shot images (see Supplementary Fig. 3 and Supplementary Methods). Since hot-spots varied from shot to shot, while e-beam and plasma structures were comparatively stable, 30-shot averaging removed probe artifacts from the data, while sharpening details of shadow/fringe patterns (Fig. 1c) for quantitative analysis. Probe angle $\theta = 8$ mrad was chosen to highlight the plasma column's expanding edge, while satisfying space constraints at the ends of the heat-pipe oven, the only optical access to the plasma column. Supplementary Figure 4 compares paths of 0.8 μm wavelength probe rays through an idealized plasma column, similar to the actual one, for $\theta = 0.008, 0.02,$ and $\pi/2$ rad. For $\theta = 0.008$ rad, probe rays sample only the plasma columns outer edge, as discussed in connection with main text (Fig. 2a–c). For $\theta = 0.02$ rad, they penetrate to the plasma axis, enabling probing of interior structures. For $\theta = \pi/2$ rad, probe rays are nearly un-deflected, rendering the plasma invisible. Transverse probe pulse width ($w_0 \approx 0.4$ cm), chosen to ensure nearly constant beam size through the heat pipe, enforced lower limit $\theta_{\min} \approx 0.007$ rad to avoid perturbing the e-beam driver with a probe injection mirror of radius $\pi w_0/2$, and upper limit $\theta_{\max} \approx 0.012$ rad to avoid clipping the probe beam on the inner aperture (radius 1.6 cm) of the oven and heat transport pipe. A single-element lens (2-inch diameter, 1 m focal length) imaged probe pulses to 12-bit CCD camera (Allied Vision Technologies Model Manta G-095B, 1292 × 734 pixels, pixel size 4.08 × 4.08 μm).

**Simulations**. OSIRIS simulations of ionization and wake formation (Fig. 3a, b) used a moving simulation box of dimensions $L_r \times L_z = 564 \times 940$ μm, divided into 0.94 × 0.94 μm cells with 12 × 12 particles per cell. The ADK model of Li ionization that these simulations use is strictly valid only up to a critical field $E_{\rm crit} = 18.7$ GV/m[45], whereas fields as high as 30 GV/m are reached near the compressed part of the bunch. Nevertheless, the first level of Li is ionized in regions where the field is ≲$E_{\rm crit}$. Thus errors from the ADK approximation are negligible. The field at the boundary of ionized and neutral Li is 13 GV/m. To model ion motion (Fig. 3e) driven by electrostatic energy stored in the nonlinear electron wake (Fig. 3c, d), we switched to a static $L_r \times L_z = 0.940 \times 3.025$ mm simulation box in the Li density plateau divided into 0.47 × 1.21 μm cells with 10 × 10 particles per cell. Impact ionization of neutrals is small on a tens-of-ps time scale, and was not included in OSIRIS simulations.

For LCODE simulations, our simulation window extended laterally out to $r = 9.4$ mm, with grid size 0.19 μm. The initial plasma consisted of $5 \times 10^4$ equal-charge macro-particles of each type within a 40 μm radius, corresponding to an average of 250 particles per cell (ppc) per species. Because of cylindrical geometry, the number ranged from ~0 at the axis to ~500 ppc at $r = 40$ μm. As the plasma expanded and impact ionization proceeded, the number increased to >1000 ppc from $r \approx 80$ μm to the plasma edge throughout most of the simulated time interval. These numbers are consistent with those used in convergence tests of other laser-plasma PIC codes[32,33]. We implemented elastic collisions via the Takizuka-Abe model[34] modified to include relativistic particles[32]. Approximate cross-sections for electron- and ion-impact ionization came from refs. [35] and [36], respectively. Modeling of two-step ionization of lithium neutrals was based on excitation and ionization cross-sections from refs. [37,38]. We developed a test to ensure that the Takizuka-Abe model implemented in LCODE accurately simulates deceleration of fast (few keV)

electrons in the energy range of interest here (see Supplementary Methods). The test compares simulated average deceleration rate $\langle \dot{p}_z \rangle$ with an analytic theoretical formula[39] (see Supplementary Eq. (2)) for different probe electron energies $E_{probe}$. Results, shown in Supplementary Fig. 5, show accurate correspondence between simulation and theory.

We simulated probe images (e.g. Fig. 1d–f) by numerically propagating a probe pulse ($\lambda = 800$ nm) at angle $\theta$ from the axis through a cylindrically symmetric column consisting of a mixture of atomic Li in $2S$ ground, and $2P$, $3S$, $3P$ excited states, plus singly-ionized Li plasma. We calculated the index of refraction $\eta_i = \eta_i(\omega, r, \Delta t)$ of atomic Li in each of the four states ($i = 1, 2, 3, 4$) at probe frequency $\omega = 2\pi c/\lambda$, radius $r$, and time delay $\Delta t$ from the Lorentz–Lorenz relation

$$3\frac{\eta_i^2 - 1}{\eta_i^2 + 2} = \frac{N_i(r, \Delta t)e^2}{\epsilon_0 m}\sum_k \frac{f_k}{-\omega^2 + i\gamma_k\omega + \omega_{0k}^2},\qquad(1)$$

where oscillator strengths $f_k$, damping factors $\gamma_k$, and resonant frequencies $\omega_{0k}$, taken from spectroscopic data in ref. [46], are known with $\pm 0.3\%$ uncertainty. The refractive index of singly-ionized Li plasma was given a Drude form

$$\eta_5 = \left[1 - \frac{\omega_p^2(r, \Delta t)}{\omega^2}\right]^{1/2}.\qquad(2)$$

LCODE simulations output partial density distributions $N_i(r, \Delta t)$ (see Fig. 5a, b) and $n_e(r, \Delta t)$ (see Fig. 4e) required to finish calculating each index contribution $\eta_i(r, \Delta t)$. We combined these, as shown in Supplementary Fig. 6, to obtain composite refractive index distributions

$$\eta(r, \Delta t) = \sum_{i=1}^{5} \eta_i(r, \Delta t)\qquad(3)$$

plotted in Fig. 4b, d, f.

## Data availability

Experimental data were generated at SLAC's FACET National User Facility. The authors declare that all data supporting the findings of this study are available within the paper and its Supplementary Information. Reasonable additional inquiries about the data should be directed to the corresponding author.

## Code availability

The authors declare that all computer codes supporting the findings of this study are fully documented within the paper, its references, and its Supplementary Information. Information about OSIRIS, including access procedures, is available at https://picksc.idre.ucla.edu/software/software-production-codes/osiris/ and http://epp.tecnico.ulisboa.pt/osiris/. Information about LCODE is available at https://lcode.info/. Reasonable additional inquiries about the codes should be directed to J.V. (on behalf of the OSIRIS consortium) or K.V.L. (LCODE).

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

## Acknowledgements

We acknowledge support from the U.S. Department of Energy (grants DE-SC0011617, DE-SC0014043 and SLAC contract DE-AC02-76SF00515), the U.S. National Science Foundation (grants PHY-1734319 and PHY-2010435), EU Horizon 2020 EuPRAXIA (grant 653782), FCT Portugal (grants PTDC/FIS-PLA/2940/2014 and SFRH/IF/01635/2015), computing time on GCS Supercomputer SuperMUC@LRZ, the Novosibirsk region government, and the Russian Foundation for Basic Research (Project 18-42-540001).

## Author contributions

R.Z. designed and installed the optical probing apparatus, with assistance from Z.L., and acquired and analyzed all experimental data. J.A., S.G., and M.L., under direction of M.J. H. and V.Y., provided logistical support that helped coordinate this experiment with other SLAC-FACET projects. T. S. and J. V. carried out OSIRIS simulations, and A.S., V.K.K. and K.V.L. LCODE simulations, of the experimental results. M.C.D. conceived, led, and acquired funding for the experiment, and wrote the paper in consultation with R.Z., M.J.H. and V.Y., and the simulation teams. All authors discussed the results and commented on the manuscript.

## Competing interests

The authors declare no competing interests.
