## [Peer Review File · Nature Communications]

Reviewers' comments:

Reviewer #1 (Remarks to the Author):

The article by Zgadzaj et al. presents measurements and simulations of an electron-beam-generated plasma wake of a plasma wakefield accelerator.

The goal of this work is to better understand how the energy deposited by the SLAC electron beam in a tenuous plasma is dissipated, which impacts the thermodynamics of the used gas cell, especially for higher repetition rates. For this, measurements from an almost-longitudinal optical refractometric probe are compared to PIC simulations of the interaction for timescales up to 1 ns. It is shown that two-stage impact ionization of the surrounding plasma needs to be included in simulations of the plasma wake on an ns timescale in order to explain the large observed radii. The paper is very well written and easy to follow. The authors explain both measurements and simulations in sufficient detail to understand the measurements and how to interpret the measured data with the help of detailed PIC simulations, as well as potential shortcomings of both the experimental and simulation tools. Overall, I am of the opinion that the results are novel and of interest to others in the community and the wider field, and recommend publication in Nature Communications.

However, I have questions about the longer-time energetics that I'd like to have answered. While the current work focuses on the first nanosecond after interaction (and does a great job explaining what is happening), you also show in Fig. 1g a measurement taken after 10 microseconds. This data point indicates that even after this long time the plasma has not recovered to an unperturbed state and is able to refract the optical probe beam out of the acceptance angle of the lens. What is causing this effect? Is that due to gas expansion or does the darker image indicate that the Li is still ionized at this time and that the remnant electrostatic fields last this long? Whatever it is, isn't that what will determine the limits on repetition rate and not so much a detailed understanding of the expanding ionization front during the first nanosecond (which apparently leaves the e-beam interaction zone at $1e6$ m/s)? Did you take measurements at longer delays and if so, at which time was the probe image unperturbed?

There is a typo in the last sentence of Methods->Probe laser and imaging system: the pixel size should be 4.08×4.08 micrometers, not mm.

Reviewer #2 (Remarks to the Author):

As an initial step in the exploration of long term dynamics associated with multi-GeV plasma wakefield acceleration, authors have extracted valuable information from a simple space and time-resolved shadowgraphic probe that reveals aftermath features following the collapse of the plasma wake generated by a highly relativistic ultrashort electron bunch traversing tenuous Li vapour. This novel work has been done with a quantitative precision that is typical of the simple probe technique used (and could be readily reproduced). In addition to the energy accounting afforded by observed dynamics, the critical value of this work includes the revealed importance of atomic structure and atomic/ionic processes in delayed secondary plasma generation over extended time intervals. There

is no doubt that this kind investigation is new and is essential in the evaluation of wake dissipation dynamics and energetics in repetition-rated accelerator and laser-driven systems. This work also exposes great opportunity for added experimental sophistication with additional probes and techniques thereby underscoring this to be a pioneering effort. The authors have highlighted the need for further similar study to reveal repetition-rated limitations for future accelerator systems of this type but also there is intimated significant potential for energy management and even control. Results given in this manuscript clearly advance the field of multi-GeV plasma wakefield accelerators by highlighting more extended behavior. This work also points to experimental enhancements in a manner that readily enables corroboration.

I make some detailed remarks about the manuscript in what follows. With these minor changes, I consider this work to be publishable in the Nature Communications journal.

In general, this manuscript is relatively-well written but it can benefit from a paragraph that summarizes all the relevant processes (in temporal order) that are discussed. In the current version, the reader has to keep track while reading along and the full picture becomes more clear by the time one reaches ~ page 4 in the description of qualitative mechanisms and simulations. A simple paragraph added near page 2 can be helpful reader guidance. Also, please confirm that prior to arrival of the electron bunch, the Li vapor is neutral (i.e. no ions). On page 2, line 5 please be sure that the formula is clearly printed to avoid misinterpretation of placing vacuum permittivity in the denominator. The calculation of the 'index -1' is an essential part of the analysis. The authors should state clearly their assumptions/symmetries used here and also give the reader some indication of the relative importance of atomic/ionic versus electronic contributions. We can surmise this to some extent in the plots of figure 4 (d) and (f). I think the space and time resolution in the 1:1 imaging system is ultimately limited by the CCD (pixel density) and ~ 1 ps in the temporal case is quite low. Can the author provide a better estimate of these coarse resolutions ? On page 3, the downstream ($z > 75$ cm) plasma lensing explanation for 'delta z' is not clear. There is also plasma lensing occurring along the entire probe. Figure 4 and 5 can be improved for readability. In figure 4, the spatial scale maximum in (a) and (b) should be reduced to about 0.150 mm in order to better reveal early time behavior. In figure 5(a), I suggest a colour scheme for distinction that is clearer (for example, the legend uses too many shades of red). Also in figure 5(a), the 'ion impact' region 4 (burgundy or purple in the colour scheme) is not clearly shown in the plot. In figure 5(c), a scale maximum near 600 ps would improve the resolution of early time behavior. Finally in figure 5 (c), it is helpful to also state in the colour legend that 'lost at walls' (light blue) applies to hot electrons in order to be more compatible with the rest of the legend.

Reviewer #3 (Remarks to the Author):

1 General comments

The manuscript "Dissipation of electron-beam-driven plasma wakes" touches on an important, yet unexplored aspect of this promising accelerator technology - the long-time evolution of and energy balance in plasma wakefields. To my knowledge, there exist no prior experimental studies of this topic on the relevant time scales.

The presented manuscript builds on relatively simple measurements of the long-term evolution of the refractive index in a plasma column heated by an incoming electron beam that excites a strong wakefield. These measurements are then compared to long time-scale simulations to support the validity of the numerical results. The potentially interesting findings presented in this paper are then extracted from the simulations, e.g. the energy balance of the dissipating plasma wave.

While I regard the presented measurements as interesting and novel, I am not convinced by the presented particle-in-cell simulations, on which the scientific interpretation rests almost exclusively. I will detail this below. Therefore, I do not believe that the claims presented in this paper are based on a rigorous scientific foundation.

In addition, the authors have unfortunately chosen to investigate the long-term evolution of a narrow beam self-ionized plasma column and not of a preionized larger diameter column. The former is considered as an inferior concept to the latter owing to an increased importance of beam head erosion and thus a decreased efficiency of the scheme. If a high-energy physics collider based on this technology is ever build, it will therefore most certainly rely on preionized, large diameter plasmas. Also most relevant laser-driven wakefield accelerators work in the large diameter plasma column regime (large compared to the transverse size of the generated wakefield). This diminishes the impact of this work since the dissipation dynamics will be different as the authors themselves confirm in their manuscript. The choice of regime here is likely motivated by the circumstance that the chosen detection method does not work as well for preionized plasmas.

Based on these considerations, I recommend to not consider this paper for publication in Nature Communications. It lacks the required high impact for the future of the field of plasma-based particle acceleration or for other sciences. While the topic itself is highly interesting, the presented study here is unconvincing. I suggest that only if the authors can unambiguously demonstrate the validity of their simulation approach, the paper should be published in a more specialized journal since the main insights gained here rest on the performed simulations.

1.1 Validity of the simulations

The chain of simulations presented in this work prompt a few questions, which are critical to clarify to consider the numerical results trustworthy. My concerns are detailed in the following.

- In the OSIRIS simulations, the authors model the ionization via the ADK model. In this context, it should be noted that the ADK model is strictly only valid up to a critical field E_{crit} and overestimates the ionization rate above E_{crit} . The authors should mention this, and evaluate if the space-charge field of the focused drive beam, which after all drives a wake in the full blowout regime, does not exceed the critical field strength. My guess is that it does. This must be considered and mentioned when it comes to explaining the discrepancy between measurement and simulation.

- My most severe concerns are with the LCODE simulations. The authors state:

”The quasi-static approximation reduces dimensionality by one, making LCODE more time-efficient for simulation larger Δt , at the cost of neglecting self-consistent evolution of drive bunch and plasma for $z > 31.9$ cm.”

The authors need to justify why neglecting the self-consistent evolution of the plasma is reasonable in this long-term evolution in the collisional regime. This is a strong assumption, which needs to be supported by physical reasons, not by numerical ones as done here.

Modelling collisions in particle-in-cell codes can be tricky. Collisional modules need to be rigorously tested as it has been done e.g. for the PIC code EPOCH [Arber et al., 2015] or for CALDER [Pérez et al., 2012]. I could find no publications or citations on the collisional module utilized in LCODE. It is absolutely mandatory to describe the collisional module in detail and to perform and show benchmarks and convergence tests. With the very limited provided information, the results of the simulations have to be considered as questionable. In particular, this impression is fueled by the low number of particles-per-cell (ppc) in the LCODE simulations. The authors state that they use a box radius of 9.4 mm, with a grid size of $0.19 \mu\text{m}$, resulting in ≈ 49500 grid cells. With 5×10^4 plasma particles this results in only a single ppc per species. For example, the benchmarking tests for EPOCH or CALDER used at minimum 30 ppc up to more than 1000 ppc to achieve convergence. [Arber et al., 2015] state in their paper that for modelling collisional effects, a high number of particles per cell is critical. The number of ppc in the used LCODE simulations may increase later on due to collisional ionization. This, however, I would not expect to be statistically relevant or change the convergence of the simulations considering the large discrepancy in ppc with other well benchmarked collisional models and implementations.

As the drawn conclusions and interpretation of the experimental data is heavily based on the simulations, it is of utmost importance to verify their convergence and reliability.

The authors list a few possible causes for the discrepancies between the measurement and the simulation. Insufficient modelling of the physics owing to a lack of particles-per-cell may actually be the most important factor. This needs to be investigated.

- Another peculiar finding indicating problems with the simulation results is shown in Fig. 5 c): There are a few bumps in the energy distributions, the strongest one at ~ 800 ps (see Figure, which shows a zoom into Fig. 5c). The step-like increase actually corresponds to a sudden jump in the distribution of the electron energy density by $\approx 18\%$. This abnormality seems nonphysical and needs to be addressed, explained and, at best, removed by the authors.

Figure 1: Snippet of Fig. 5 c)

1.2 Minor questions regarding the experiment

- The authors do not quantitatively state the temporal resolution in their measurements. They only claim their choice of propagation angle between laser and electron beam comes at a "mild cost" in temporal resolution (page 2, last line). What is this cost exactly? Does the mismatch between laser group velocity and electron beam velocity play a role?

- I do not agree with the statement on page 3, first paragraph: "Use of an e-bunch driver eliminated strong forward-directed supercontinuum that a laser-driven nonlinear wake generates, which would saturate probe detectors in this near co-propagating pump-probe geometry." There, in principle, is the possibility to separate longitudinal probe and LWFA laser by polarization. Therefore, this statement in its generality seems to be too strong.

References

- TD Arber, Keith Bennett, CS Brady, A Lawrence-Douglas, MG Ramsay, NJ Sircombe, P Gillies, RG Evans, Holger Schmitz, AR Bell, et al. Contemporary particle-in-cell approach to laser-plasma modelling. *Plasma Physics and Controlled Fusion*, 57(11):113001, 2015.
- F Pérez, L Gremillet, A Decoster, M Drouin, and E Lefebvre. Improved modeling of relativistic collisions and collisional ionization in particle-in-cell codes. *Physics of Plasmas*, 19(8):083104, 2012.

Key to type: 11 pt. *italic type: reviewer comments*

11 pt. standard type: our responses to Rev. 1 (green), Rev. 2 (red), Rev. 3 (blue)

Boldface type: description of manuscript changes in response to comments

Reference numbers in responses below are those of the revised manuscript.

Response to Reviewer #1.

Reviewer #1 comment 1: *The article by Zgadzaj et al. presents measurements and simulations of an electron-beam-generated plasma wake of a plasma wakefield accelerator. The goal of this work is to better understand how the energy deposited by the SLAC electron beam in a tenuous plasma is dissipated, which impacts the thermodynamics of the used gas cell, especially for higher repetition rates. For this, measurements from an almost-longitudinal optical refractometric probe are compared to PIC simulations of the interaction for timescales up to 1 ns. It is shown that two-stage impact ionization of the surrounding plasma needs to be included in simulations of the plasma wake on an ns timescale in order to explain the large observed radii.*

The paper is very well written and easy to follow. The authors explain both measurements and simulations in sufficient detail to understand the measurements and how to interpret the measured data with the help of detailed PIC simulations, as well as potential shortcomings of both the experimental and simulation tools. Overall, I am the opinion that the results are novel and of interest to others in the community and the wider field, and recommend publication in Nature Communications.

Our response: We appreciate this positive assessment.

Reviewer #1 comment 2: *However, I have questions about the longer-time energetics that I'd like to have answered. While the current work focuses on the first nanosecond after interaction (and does a great job explaining what is happening), you also show in Fig. 1g a measurement taken after 10 microseconds. This data point indicates that even after this long time the plasma has not recovered to an unperturbed state and is able to refract the optical probe beam out of the acceptance angle of the lens. What is causing this effect? Is that due to gas expansion or does the darker image indicate that the Li is still ionized at this time and that the remnant electrostatic fields last this long? Whatever it is, isn't that what will determine the limits on repetition rate and not so much a detailed understanding of the expanding ionization front during the first nanosecond (which apparently leaves the e-beam interaction zone at 1e6 m/s)? Did you take measurements at longer delays and if so, at which time was the probe image unperturbed?*

Our response: Yes, indeed, the state of the plasma at 10 μ s (and beyond) is what will determine repetition rate. That is why we included Fig. 1g in the manuscript. Our manuscript's focus on the 1st ns is the result of three considerations:

(1) *To understand the state of the plasma at 10 μ s (and beyond), one must first understand the 1st ns of plasma dynamics. Our results in Fig. 5b,c show complicated transformations among different collisional*

processes (Fig. 5b), accompanied by a complicated transfer of energy from fields to electrons to ions (Fig 5c), during the first ~ 300 ps. Collisional processes and energy partitioning then reach steady state, with outwardly streaming ions having acquired $\sim 85\%$ of the original wake energy, which persists for the next ~ 900 ps, and presumably well beyond. Establishment of this steady state during the 1st ns enables broad predictability of the future state of the plasma, and is a prerequisite to understanding that future state.

(2) *We can simulate the 1st ns of plasma dynamics; we cannot simulate the first 10 μ s.* Even simulating 1 ns of plasma dynamics represents the frontier of computational plasma science. Comparing these simulations to our data enabled us to understand the first ns of plasma evolution in unprecedented detail, and even to conclude, based on the “steady state” in “plasma energy and its partitioning among plasma species” reached for $\Delta t > 300$ ps “...that the $\sim 10^6$ m/s expansion continues unabated well beyond $\Delta t \approx 1.3$ ns” (p. 6). The first nanosecond therefore comprised a new and publishable scientific story.

(3) *Experimental run time at a national laboratory is limited; full understanding of the plasma’s μ s and ms dynamics will take much longer than the current study.* We were not allocated sufficient time during FACET-I to explore these long-time dynamics in detail. We have not yet found a Δt “at which the probe image is unperturbed”. We hope to do so in future runs. But at present, in view of sparse data and lack of supporting simulations, we simply cannot yet answer the reviewer’s interesting questions about plasma dynamics at these delays, and we elected not to indulge in speculation in this paper. At the same time, we felt it was irresponsible to omit mention of our limited μ s-scale findings entirely, simply in order to avoid this line of questioning. Thus we presented Fig. 1g, along with the related observation that “when ... we increased repetition rate to ~ 10 Hz, oven temperature typically rose tens of degrees within minutes...” as empirical findings. As Reviewer #2 correctly states, we have taken “...an initial step in the exploration of long-term dynamics associated with multi-GeV plasma wakefield acceleration”.

Reviewer #1 comment 3: *There is a typo in the last sentence of Methods->Probe laser and imaging system: the pixel size should be 4.08x4.08 micrometers, not mm.*

Our response: Thanks. **We corrected it.**

Response to Reviewer #2.

Reviewer #2 comment 1: *As an initial step in the exploration of long term dynamics associated with multi-GeV plasma wakefield acceleration, authors have extracted valuable information from a simple space and time-resolved shadowgraphic probe that reveals aftermath features following the collapse of the plasma wake generated by a highly relativistic ultrashort electron bunch traversing tenuous Li vapour. This novel work has been done with a quantitative precision that is typical of the simple probe technique used (and could be readily reproduced). In addition to the energy accounting afforded by observed dynamics, the critical value of this work includes the revealed importance of atomic structure and atomic/ionic processes in delayed secondary plasma generation over extended time intervals. There is no doubt that this kind investigation is new and is essential in the evaluation of wake dissipation dynamics and energetics in repetition-rated accelerator and laser-driven systems. This work also exposes great opportunity for added experimental sophistication with additional probes and techniques thereby underscoring this to be a pioneering effort. The authors have highlighted the need for further similar study to reveal repetition-rated limitations for future accelerator systems of this type but also there is intimated significant potential for energy management and even control. Results given in this manuscript clearly advance the field of multi-GeV plasma wakefield accelerators by highlighting more extended behavior. This work also points to experimental enhancements in a manner that readily enables corroboration. I make some detailed remarks about the manuscript in what follows. With these minor changes, I consider this work to be publishable in the Nature Communications journal.*

Our response: We appreciate this positive assessment.

Reviewer #2 comment 2: *In general, this manuscript is relatively-well written but it can benefit from a paragraph that summarizes all the relevant processes (in temporal order) that are discussed. In the current version, the reader has to keep track while reading along and the full picture becomes more clear by the time one reaches ~page 4 in the description of qualitative mechanisms and simulations. A simple paragraph added near page 2 can be helpful reader guidance. Also, please confirm that prior to arrival of the electron bunch, the Li vapor is neutral (i.e. no ions).*

Our response: We expanded the last paragraph of the Introduction (p.2) to summarize the chronological sequence of relevant processes, as the reviewer suggests. Therein we confirm explicitly that the lithium vapor is neutral before the drive bunch arrives. Alterations are in bold-faced type:

Here we present ps-time-resolved optical shadowgraphic measurements of meter-length ion channels that emerge from broken, highly nonlinear plasma wakes **generated in self-ionized, initially neutral lithium vapor. As the drive bunch propagates ~0.3m into the vapor, its center and trailing edge self-focus to ~100x their original density, enabling them to field-ionize a wide plasma column and to drive a strongly nonlinear plasma wake within its boundaries. The drive bunch then reaches steady state, and continues generating this nonlinear wake over the next meter. A diagnostic optical pulse probes the expanding plasma column at a fixed longitudinal location z within this steady-state region at time delays $0 < \Delta t < 1.5$ ns in 0.1 ns intervals, and more coarsely out to 10 μ s.** These observations serve as a calorimeter that determines the fraction of the initial wake energy that the plasma column retains after the wake breaks. **Simulations reveal that the initial wake transfers energy into the expanding plasma column via the following sequence of events: the driver expels some plasma electrons into surrounding gas; the initial wake breaks, retaining radial electric fields that propel ions outward at tens of keV while**

escorting electrons; outwardly streaming electrons and ions impact-ionize and excite surrounding neutral lithium, expanding plasma volume several hundred-fold. Benchmarking simulated plasma expansion against measurements quantifies energy retention in the plasma column and elucidates the physical mechanisms that drive its expansion.

Reviewer #2 comment 3: On page 2, line 5 please be sure that the formula is clearly printed to avoid misinterpretation of placing vacuum permittivity in the denominator.

Our response: We corrected this ambiguity in the manuscript with the following change in the text: “ $|E|^2/(2\epsilon_0)$ ”

Reviewer #2 comment 4: The calculation of the ‘index -1’ is an essential part of the analysis. The authors should state clearly their assumptions/symmetries used here and also give the reader some indication of the relative importance of atomic/ionic versus electronic contributions. We can surmise this to some extent in the plots of figure 4 (d) and (f).

Our response: To address the reviewer’s request, we added two plots (Fig. S6a,b) in the Supplementary Material (new Sec. 6) that show an example of (a) simulated radial density distribution of five constituent states/species considered --- neutral lithium in the 2s, 2p, 3s and 3p states, and singly ionized lithium --- at $\Delta t = 400\text{ps}$, (b) their corresponding contributions to the refractive indexes. The brown curve and shaded area show the overall effective refractive index. The refractive indices η_i ($i = 1-4$) of the four neutral atom states are obtained from the Lorentz-Lorenz relation (Eq. 2) based on tabulated polarizabilities of lithium in each state. The plasma refractive index η_5 is calculated from the plasma density (Eq. 3). The effective refractive index depends on partial fractions of the five states/species, which vary with time and radius, leading to a complex time-varying evolution of the medium’s refractive index. In most cases plasma refractive index η_5 dominates in the core of the plasma column, while the neutral 2S state refractive index η_1 dominates at larger radii. The transition between these evolves with time. The relative importance of the 2p, 3s, and 3p states is lower, but becomes significant at radii near the transition from ionized to partly ionized lithium.

Fig. S6. (a) LCODE result for the radial distribution of five constituent states of lithium at 400ps after passage of electron beam. (b) refractive index distributions at 800nm, corresponding to the densities shown in (a), including the resultant effective refractive index. These results form the basis of the families of plots shown in Fig. 4.

Reviewer #2 comment 5: *I think the space and time resolution in the 1:1 imaging system is ultimately limited by the CCD (pixel density) and ~ 1 ps in the temporal case is quite low. Can the author provide a better estimate of these coarse resolutions?*

Our response: Pixel size is not a limiting factor in the imaging system. Several factors determine overall temporal resolution: (1) probe pulse duration; (2) geometric walk-off due to the oblique path of the probe through the plasma column; (3) optical resolution of the imaging system. We evaluate the “cost” of each in turn.

(1) Probe duration is ~ 75 fs, and contributes as much to temporal resolution.

(2) Geometric walk-off depends on transit angle $\theta = 8$ mrad (0.46°), plasma column density profile $n_e(r)$ (varies with Δt), and the difference between probe group velocity and e-bunch velocity. The combined effect on temporal resolution becomes comparable to that of probe duration for plasma column radii $r \sim 0.5$ mm, and is limited to ~ 100 fs for all data shown. Thus it is negligible compared to ns time scales discussed in this work. This is the “mild cost” mentioned above.

(3) Transverse imaging resolution limits temporal resolution most severely. The necessity of imaging the plasma column through the end of a long, narrow pipe dictates effective $f/\# \sim 50$ at the upstream ($f/\# \sim 25$ at the downstream) end of each image, corresponding to spatial resolution limit ~ 50 μm (~ 25 μm). This in turn leads to blurring of temporal information of order $\sim (25$ μm to 50 $\mu\text{m}) / (0.008\text{rad}) / (3 \times 10^8\text{m/s}) \approx 10\text{ps}$ to 20ps , about 2 orders of magnitude greater than the effect of the probe duration and geometric walk-off.

Thus the overall temporal resolution of the presented measurements near the center of the images, where we report the column’s radial profile, is ~ 15 ps, much smaller than the 100ps time steps in the data. In the revised manuscript, **we added the boldfaced passages below to state this temporal resolution quantitatively** at the end of the section “Results/ Generation of nonlinear wakes”:

Nevertheless, ionization induces a large change in the vapor's refractive index η , enabling us to detect the ionization front **with ~ 15 ps time and < 40 μm space resolution (both limited primarily by imaging resolution)**, without perturbing overall energy transport dynamics significantly.

Reviewer #2 comment 6: *On page 3, the downstream ($z > 75$ cm) plasma lensing explanation for ‘delta z ’ is not clear. There is also plasma lensing occurring along the entire probe.*

Our response: Indeed plasma lensing occurs along the entire path where the probe light passes through the column. Our explanation was poorly worded, and we thank the reviewer for pointing this out. **We revised the explanation with the boldfaced passages below:**

At small Δt the vertex of the parabola **appears to be located at $z = 75$ cm** [see 100ps images in Fig. 1(c)-(f)], i.e. **the position in the images where the object plane intersects the plasma column. In fact, the vertex is shifted slightly to $z \approx 75\text{cm} + \Delta z$.** As the plasma column widens, Δz increases [see 100ps to 1200ps images in Fig. 1(c) and (f)]. **This is the result of increasing refraction in the cylindrical plasma column.**

Reviewer #2 comment 7: Figure 4 and 5 can be improved for readability.

In figure 4, the spatial scale maximum in (a) and (b) should be reduced to about 0.150 mm in order to better reveal early time behavior.

In figure 5(a), I suggest a colour scheme for distinction that is clearer (for example, the legend uses too many shades of red).

Also in figure 5(a), the 'ion impact' region 4 (burgundy or purple in the colour scheme) is not clearly shown in the plot.

In figure 5(c), a scale maximum near 600 ps would improve the resolution of early time behavior.

Finally in figure 5 (c), it is helpful to also state in the colour legend that 'lost at walls' (light blue) applies to hot electrons in order to be more compatible with the rest of the legend.

Our response: We appreciate the suggestions, all of which we have adopted.

New Figure 5.

Old Figure 5.

Response to Reviewer #3. Reviewer #3 divided comments into three groups: 1. General Comments, 2. Validity of the simulations, and 3. Minor questions regarding the experiment.

Reviewer #3: Group 1. General comments

Reviewer #3 comment 1.1: *The manuscript "Dissipation of electron-beam-driven plasma wakes" touches on an important, yet unexplored aspect of this promising accelerator technology - the long-time evolution of and energy balance in plasma wakefields. To my knowledge, there exist no prior experimental studies of this topic on the relevant time scales.*

The presented manuscript builds on relatively simple measurements of the long-term evolution of the refractive index in a plasma column heated by an incoming electron beam that excites a strong wakefield. These measurements are then compared to long time-scale simulations to support the validity of the numerical results. The potentially interesting findings presented in this paper are then extracted from the simulations, e.g. the energy balance of the dissipating plasma wave.

Our response: We appreciate the reviewer's recognition of the importance of the subject studied in this work and its novelty.

Reviewer #3 comment 1.2: *While I regard the presented measurements as interesting and novel, I am not convinced by the presented particle-in-cell simulations, on which the scientific interpretation rests almost exclusively. I will detail this below. Therefore, I do not believe that the claims presented in this paper are based on a rigorous scientific foundation.*

Our response: We appreciate the reviewer's thorough study of the manuscript. Below we respond in detail to each specific criticism. The reviewer's main concerns about scientific rigor appear to be based on misunderstandings, which we have endeavored to resolve.

Reviewer #3 comment 1.3a: *In addition, the authors have unfortunately chosen to investigate the long-term evolution of a narrow beam self-ionized plasma column and not of a preionized larger diameter column.*

Our response: As discussed below, the reviewer may have misunderstood how "narrow" the self-ionized plasma column is: its radius ($R_p \approx 40 \mu\text{m}$, see Fig. 3b) is twice the radius of the wake ($R_w = \lambda_p/2\pi = 20 \mu\text{m}$, see also Fig. 3b) generated within it. With this initial condition, wake dissipation dynamics differ negligibly between self- and realistic finite-radius pre-ionized plasmas of equivalent R_p/R_w , contrary to the reviewer's premise. Thus results presented are representative of dissipation dynamics of an equivalent wake generated in a pre-ionized plasma column of radius $R_p \approx 2R_w$. The *long-term* ($\Delta t \gg 100$ ps) dynamics --- the main interest here --- are also representative of those of an equivalent wake in wider, but *finite*, pre-ionized plasmas ($2 \lesssim R_p/R_w, \lesssim 5$). When comparing different, but finite, R_p/R_w , significant differences arise only in the plasma's *short-term* (tens of ps) evolution, i.e. on the time scale for outwardly streaming energy to reach R_p . Hence there is nothing "unfortunate" about our choice. On the contrary, it is the best choice for a first experimental investigation of long-term plasma evolution following excitation of a nonlinear wake, because the long-term evolution is easier to observe in a self-

ionized plasma column, and the results apply equally to wakes in self- and pre-ionized plasma columns of similar dimensions.

Reviewer #3 comment 1.3b: *The former is considered as an inferior concept to the latter owing to an increased importance of beam head erosion and thus a decreased efficiency of the scheme. If a high-energy physics collider based on this technology is ever build, it will therefore most certainly rely on preionized, large diameter plasmas.*

Our response: The range $2 \lesssim R_p/R_w \lesssim 5$ which our results represent is the most practical configuration for PWFA colliders because R_p is large enough to avoid head erosion (in the equivalent pre-ionized case), but small enough to avoid prohibitive energy cost. Pre-ionized columns of $5 < R_p/R_w < \infty$ generated by wallplug-inefficient lasers, and replicated over tens to hundreds of PWFA stages [Adli, SLAC-PUB-15426, ArXiv: 1308.1145], would be the facility’s dominant energy consumer. Moreover, non-laser-based pre-ionization methods, such as electrical discharge channels [Spence, PRE63, 15401 (2001)], do not work in the plasma-density ($10^{16} < n_e < 10^{17} \text{ cm}^{-3}$) or length ($\sim 1 \text{ m}$) ranges of interest for PWFAs. They have been used successfully only in connection with LWFAs at higher n_e and shorter length [Gonsalves (2019)]. Recent efforts to generate pre-ionized channels of density down to $n_e \sim 10^{17} \text{ cm}^{-3}$ re-introduce inefficient lasers, either in conjunction with [Bobrova, Phys. Plasmas 20, 020703 (2013)], or independently of [Lemos, Phys. Plasmas 20, 103109 (2013)], discharge channels. Recent literature on pre-ionizing and shaping low- n_e , meter-length plasma columns for PWFAs invariably focus on laser-based methods [Gessner, *Nat. Commun.* 7, 11785 (2016)].

We are not promoting self-ionized PWFA collider technology, nor any specific application, but rather elucidating fundamental wake dissipation physics that is equally relevant to both self- and finite-radius pre-ionized PWFAs. As long as the initial field- or pre-ionized plasma column size is “large compared to the transverse size of the generated wakefield” (to borrow a phrase from the reviewer’s next comment) --- a condition that is satisfied in our experiment --- the drive bunch creates initial wakes of very similar amplitude, structure and stored energy in the two cases, as the simulations of Deng et al., *Phys. Rev. E* 68, 047401 (2003) [new Ref. 21] showed years ago. In fact Fig. 2 of Deng’s paper (reproduced below) shows that our lithium density $n_a = 8 \times 10^{16} \text{ cm}^{-3}$ is close to the “cross-over” point at which a drive beam comparable to ours generates wakes of nearly identical amplitude in self- and pre-ionized plasma:

Fig. R1. Simulated wake amplitude E_z , normalized to cold non-relativistic wave-breaking field $E_p = mc\omega_p/e = 11.4 \text{ GV/m}$ at reference density $n_0 = 1.4 \times 10^{16} \text{ cm}^{-3}$ vs. lithium density n normalized to n_0 , created by 50 GeV e-bunch driver with 2×10^{10} electrons, $\sigma_r = 20 \mu\text{m}$, $\sigma_z = 63 \mu\text{m}$ incident upon neutral or pre-ionized gas [from Deng 2003]. Red arrow denotes density for our experiments.

True, beam head erosion causes the wake to decay faster with propagation distance z in the field-ionized case [Li, AIP Conf. Proc. 1507, 582 (2012), new Ref. 22]. But this has no bearing on our results for two reasons:

(1) *Our experimental and simulation results (Figs. 2-5) report transverse plasma dynamics at fixed z.* Thus the question of z-evolution does not enter the discussion. The fundamental ns-scale physics of wake dissipation at this fixed z in self- and pre-ionized configurations differ only in that, in the former case, outwardly streaming electrons and ions (which carry most of the wake's energy) must expend some of that energy ionizing neutral lithium atoms. But as our results show, they expend < 1% of their energy in so doing within the space/time scale of our experiments and simulations. Thus ns-scale wake dissipation dynamics are essentially identical in self- and finite-radius pre-ionized cases, and our results are equally applicable to both. We added the bold-faced passage below to the 2nd paragraph of the section "Results/Generation of nonlinear wakes" to express and justify this point:

Particle-in-cell (PIC) simulations discussed below showed that the SLAC *e*-bunches singly field-ionized the Li vapor over its entire length out to initial radius $r(0) \approx 40\mu\text{m}$ and electron density $n_e = n_a$, and drove a strongly nonlinear electron density wake consisting of a train of nearly fully blown-out cavities of radius $\tilde{\lambda}_p = 20\ \mu\text{m}$, propagating at $\sim c$ along the axis of the resulting plasma. **Thus the initial plasma column was wide enough to fully support the generated wake. Under these conditions, the wake closely resembles the corresponding wake that the *e*-bunch would form in pre-ionized plasma of similar n_e (Ref. 21). The main differences are that, in the self-ionized case, the wake forms further back in the drive bunch profile²¹ and decays more rapidly along *z* due to head erosion^{22,23}. For our conditions, ~5% of the drive *e*-bunch erodes over 1 m of propagation (see Supplementary Material, Sec. 2). Neither difference has any bearing on fixed-*z*, ns-scale, transverse plasma expansion dynamics of interest here. These dynamics are determined by the amplitude, structure and stored energy of the initial wake at the longitudinal location *z* at which we measure and simulate them, regardless of whether that wake formed in self- or pre-ionized plasma.**

(2) *Head erosion is weak under our conditions.* From simulations, the number of μm that the field-ionization front slips backward per meter of *e*-beam propagation (a measure of head erosion rate) is [22]:

$$V[\mu\text{m}/\text{m}] = (3.66 \times 10^4) (E_i [\text{eV}])^{1.73} \gamma^{-1} \varepsilon_n [\text{mm-mrad}] (I [\text{kA}])^{-3/2} \approx 2.5\ \mu\text{m}/\text{m},$$

where we obtained the numerical result using parameters $E_i = 5.39\ \text{eV}$ (ionization energy of Li), $\gamma = 4 \times 10^4$ (Lorentz factor of 20 GeV drive beam), $\varepsilon_n \approx 5\ \text{mm-mrad}$ (normalized emittance of FACET-I beam at IP) and $I \approx 10\ \text{kA}$ (peak current of 2 nC, $\sigma_z = 55\ \mu\text{m}$ FACET-I beam). Since the initial beam length is $\sigma_z = 55\ \mu\text{m}$, estimated erosion is < 5% over the entire 1 m interaction length. This can be considered negligible for our conditions. **We added a description of this estimate to Sec. 2 of the revised Supplementary Material.**

Reviewer #3 comment 1.3c: *Also most relevant laser-driven wakefield accelerators work in the large diameter plasma column regime (large compared to the transverse size of the generated wakefield).*

Our response: We do not know what the reviewer means by a "relevant" LWFA. GeV LWFAs both with [e.g. Leemans, PRL 113, 245002 (2014); Gonsalves, PRL 122, 084801 (2019)] and without [Clayton, PRL 105, 105003 (2010); Wang, Nat. Commun. 4, 1988 (2013); Kim, PRL 111, 165002 (2013)] a pre-ionized plasma column have been reported. If "relevant" means "plasma column large compared to the transverse size of the generated wakefield", as the comment's wording appear to suggest, then all of the above examples, in fact all LWFAs of which we are aware, are relevant. But then, as noted in our previous response, our current experiment also satisfies the same "relevance" criterion. Thus we do not understand what point the reviewer is trying to make with this remark.

We also fail to see why the reviewer is comparing LWFAs and PWFAs in this context in the first place. [Leemans et al., 2014] and [Gonsalves et al., 2019] did not pre-ionize the plasma in order to avoid head erosion of the drive laser pulse by ionization. Such erosion is negligible and confined to the pulse's far leading edge. The purpose of pre-ionization was rather to create a waveguide to counter laser diffraction, thereby enabling the drive laser pulse to propagate to its depletion length. Electron drive bunches, on the other hand, diffract negligibly over their depletion length. Thus the reviewer's statement about use of wide pre-ionized plasma in some LWFAs is irrelevant to the question of self- vs. pre-ionized plasma in PWFAs.

Reviewer #3 comment 1.3d: *This diminishes the impact of this work since the dissipation dynamics will be different as the authors themselves confirm in their manuscript.*

Our response: We respectfully disagree with this characterization of our manuscript. As pointed out in our previous responses, *long-term* ($\Delta t > 100$ ps) wake dissipation dynamics are nearly identical for self- vs. finite-radius pre-ionized plasma --- as long as the reviewer's criterion "plasma column large compared to transverse size of the generated wakefield" is satisfied --- directly contrary to the reviewer's assertion. We stated this in the following two passages:

- 1) p. 2, 1st paragraph under Results: "...we retained the Li vapor blanket as an *in-situ* medium for recording energy transport out of the directly-excited wake. As discussed below, outwardly streaming ions and electrons carry away most of the wake's energy, and *expend < 1% of it ionizing Li within the first ~ 1 ns.* Nevertheless, ionization induces a large change in the vapor's refractive index η , enabling us to detect the ionization front.... *without perturbing overall energy transport dynamics significantly.*" [italics added for emphasis.] n.b. the key points in italics are moved to the next paragraph in the revised mss.
- 2) p. 6, end of penultimate paragraph: "Here, plasma energy and its partitioning among plasma species reach steady state for $\Delta t > 300$ ps, and change negligibly through the end of the simulation 3 run at $\Delta t = 1300$ ps. This validates the experiment's premise that the Li blanket records energy transport via ionization *without noticeably depleting the energy of the ionizing radiation.*" [italics added for emphasis]

These passages clearly state that ionization negligibly perturbs energy transport outward from the broken wake, directly contrary to the reviewer's statement.

The reviewer appears to have reached the opposite conclusion by misinterpreting two alleged statements to the contrary by "the authors themselves":

- 3) p. 4, discussion of OSIRIS simulation results: "These features [of the ion density structure at $\Delta t = 40$ ps] contrast with observations of Ref. 5, where a region almost void of plasma formed around the on axis peak, surrounded by a thin cylindrically symmetric plasma sheath. This difference arises because in our case the initial plasma radius is less than $\lambda_p \approx 120$ μm , whereas in Ref. 5 the 75x denser plasma was pre-ionized beyond $\lambda_p \approx 14$ μm . Short-term ($\Delta t < 40$ ps) evolution of the plasma thus differed from that investigated here."

In this passage, we unfortunately compared the plasma column radius to “ λ_p ”, instead of to the wakefield radius $\tilde{\lambda}_p = \lambda_p/2\pi = 20 \mu\text{m}$ that is shown in Fig. 3b. Thus our correct statement “the initial plasma radius is less than $\lambda_p \approx 120 \mu\text{m}$ ” probably unwittingly fueled the misunderstanding that it was 3x smaller than the wake radius, rather than 2x larger. We regret this confusion. In *both* cases described, plasma column radius *exceeded* wake radius, just by different fractions. Moreover, as stated, we were describing minor differences in SHORT-TERM ($\Delta t < 40 \text{ ps}$) dynamics between self- and pre-ionized plasma. We did so simply to place our work in the context of recent prior work. However, these remarks do not apply to LONG-TERM dynamics, which were not studied in Ref. 5, but are the main subject of our paper. **In the revised manuscript, we deleted the above passage**, since the short-term differences described are incidental to our main topic, the comparison is complicated and confusing, and we wish to avoid similar misinterpretation by other readers.

- 4) p. 6, discussion of plasma energy budget (Fig. 5): “Overall 90% of the initially deposited energy remains in or near the plasma column. This contrasts with radially unbounded plasmas, in which fast electrons escape the heated region freely, cooling it to sub-keV temperatures within a few hundred wake periods¹⁵.”

In this passage, we were comparing our long-term simulation results to simulation results of Ref. 15 for *completely unbounded* plasma. Unbounded plasma, however, is impractical because of the enormous energy cost of producing it. Rapid escape of fast electrons does not affect the validity of the main results of Ref. 15, which was concerned with short-term ion wake formation. However, that rapid escape cannot be extrapolated to our parameters, nor to the practical case PWFAs in finite-radius plasma columns. In the revised manuscript, we added the bold-faced passages below to clarify the comparison with the infinite radius idealization:

“Overall 90% of the initially deposited energy remains in or near the plasma column. **Similar confinement is expected for wakes in finite-radius pre-ionized columns.** This contrasts with radially unbounded plasmas, in which fast electrons escape the heated region freely, cooling it to sub-keV temperatures within a few hundred wake periods¹⁵. **Unbounded plasma, however, is impractical for colliders because of the enormous energy cost of producing it over hundreds of accelerator stages.**”

Reviewer #3 comment 1.3e: *The choice of regime here is likely motivated by the circumstance that the chosen detection method does not work as well for pre-ionized plasmas.*

Our response: The reviewer’s speculation about motivation is not correct. The primary motivation for our choice of regime was that its long-term dissipation dynamics were similar to, and representative of, those of practical PWFA schemes, including self-ionized and finite-radius pre-ionized plasma. The “chosen detection method” is applicable to both regimes. Practical considerations provided secondary motivation only for which regime to study FIRST, as they do in any experiment. Probing finite-radius pre-ionized plasma would have required us to divide pulses from the single available laser at FACET into a strong pre-ionizing pulse and a weak probe, with a long adjustable delay between them. Space and infrastructure to implement the necessary pulse division were not available at FACET-I for experiments

presented here. By probing self-ionized plasma, we could devote the single available laser to the single purpose of probing this plasma.

Reviewer #3 comment 1.4a: *Based on these considerations, I recommend to not consider this paper for publication in Nature Communications. It lacks the required high impact for the future of the field of plasma-based particle acceleration or for other sciences. While the topic itself is highly interesting, the presented study here is unconvincing.*

Our response: As discussed in our response to comment 1.3, “these considerations” consist of a false premise [presumed large difference between long-term dissipation dynamics of wakes generated in self- vs. pre-ionized plasma (comment 1.3a,b,d)], an irrelevant analogy to LWFAs (comment 1.3c), a misreading of our manuscript (comment 1.3d), and an incorrect speculation about our motivation for choosing a detection method and an incorrect evaluation of that method’s applicability to wakes generated in pre-ionized plasma (comment 1.3e). “These considerations” do not constitute a valid basis for judging the validity or impact of our paper.

Reviewer #3 comment 1.4b: *I suggest that only if the authors can unambiguously demonstrate the validity of their simulation approach, the paper should be published in a more specialized journal since the main insights gained here rest on the performed simulations.*

Our response: “The validity of the simulation approach” is a separate issue from the “considerations” raised in Comment 1.3. We therefore address the reviewer’s concerns about the simulations in our response to the next group of comments.

Reviewer #3: Group 2. Validity of the simulations

The chain of simulations presented in this work prompt a few questions, which are critical to clarify to consider the numerical results trustworthy. My concerns are detailed in the following.

Reviewer #3 comment 2.1: *In the OSIRIS simulations, the authors model the ionization via the ADK model. In this context, it should be noted that the ADK model is strictly only valid up to a critical field E_{crit} and overestimates the ionization rate above E_{crit} . The authors should mention this, and evaluate if the space-charge field of the focused drive beam, which after all drives a wake in the full blowout regime, does not exceed the critical field strength. My guess is that it does. This must be considered and mentioned when it comes to explaining the discrepancy between measurement and simulation.*

Our response: We thank the reviewer for pointing out this limitation. **We addressed it by adding the following passage to Methods/Simulations:**

The ADK model of Li ionization that these simulations use is strictly valid only up to a critical field $E_{crit} \approx 18.7$ GV/m [Bruhwiler et al., Phys. Plasmas **10, 2022 (2003)], whereas fields reach 30 GV/m near the compressed part of the bunch. Nevertheless, the first level of Li is ionized in regions where the field is $\sim E_{crit}$. Thus errors from the ADK approximation are negligible. The field at the boundary of ionized and neutral lithium is ~ 13 GV/m.**

The errors resulting from the ADK approximation mostly affect the longitudinal location of the ionization front. The transition region between partly and fully ionized plasma is negligibly small. Therefore, the main uncertainty is limited to what fraction of the drive bunch falls within the ionized plasma, and therefore what fraction participates in the generation of the plasma wave. Consequently, the ADK approximation introduces only a small quantitative error.

Reviewer #3 comment 2.2: *My most severe concerns are with the LCODE simulations.*

The authors state: "The quasi-static approximation reduces dimensionality by one, making LCODE more time-efficient for simulation larger Δt , at the cost of neglecting self-consistent evolution of drive bunch and plasma for $z > 31.9$ cm." The authors need to justify why neglecting the self-consistent evolution of the plasma is reasonable in this long-term evolution in the collisional regime. This is a strong assumption, which needs to be supported by physical reasons, not by numerical ones as done here.

Our response: This is a misunderstanding resulting from a poor choice of wording on our part. We thank the reviewer for pointing it out. The LCODE simulation is in fact fully self-consistent. **Our phrase "at the cost of neglecting self-consistent evolution" was not correct, and we have removed it (p. 4, left column, paragraph 3).** The intended meaning was simply that once the electron bunch passes the plasma cross-section at $z = 31.9$ cm, plasma dynamics *at this cross-section* do not depend on later evolution of the electron drive bunch in the downstream plasma, i.e for $z > 31.9$ cm. This allows LCODE to simulate long-term evolution at a fixed location z efficiently, making simulations of the 1.3 ns evolution of the plasma column feasible. **The revised passage now reads:**

To simulate long-term plasma evolution, we input the compressed e -bunch and plasma profiles from the OSIRIS simulation output at $z = 31.9$ cm into the quasi-static axisymmetric LCODE²⁷ as initial conditions. **LCODE then simulated plasma dynamics at this fixed z , which do not depend on continuing evolution of the drive bunch in the downstream plasma (i.e. at $z > 31.9$ cm). Consequently, the bunch evolution no longer needed to be tracked, enabling time-efficient simulation of long-term plasma dynamics out to $\Delta t = 1.3$ ns.**

We also revised the first sentence of the subsection "Simulations of plasma expansion" by adding the bold-faced phrase below:

To understand plasma expansion quantitatively, we carried out PIC simulations of Li plasma dynamics out to $\Delta t \sim 1.3$ ns, **using two complementary PIC codes, OSIRIS and LCODE.**

Reviewer #3 comment 2.3: *Modelling collisions in particle-in-cell codes can be tricky. Collisional modules need to be rigorously tested as it has been done e.g. for the PIC code EPOCH [Arber et al., 2015] or for CALDER [Pérez et al., 2012]. I could find no publications or citations on the collisional module utilized in LCODE. It is absolutely mandatory to describe the collisional module in detail and to perform and show benchmarks and convergence tests.*

Our response: Indeed, our ionization and collision module is new in LCODE, and its detailed description is not published. However, we do not claim any novelty in this new addition to LCODE. Instead, we use already well-established collisional/ionization schemes to avoid precisely the problem that the reviewer describes. The Takisuka-Abe model need not be tested any more, after 40 years of successful

implementation in various codes. In fact, the existence of extensive tests, such as the one (Pérez et al. 2012) that the reviewer cites, effectively prevents us from publishing new test results. Moreover, the conditions simulated in the current work do not fall into the extreme parameter ranges (e.g. highly relativistic collisions, high density) for which these schemes were designed, further reducing potential issues with imprecision that concern the reviewer.

However, one effect that is important to us --- collisional deceleration of fast electrons --- was not fully addressed in available publications. Therefore we developed the following special test to convince ourselves that the Takizuka-Abe model indeed works well for our parameters of interest. A beam of probe electrons with equal momentum $(0, 0, p_z)$ and energy E_{probe} interacts with a medium of Maxwellian electrons with temperature T and density n . The initial deceleration rate is given by the following analytic formula [B. A. Trubnikov, "Particle Interactions in a Fully Ionized Plasma," *Reviews of Plasma Physics* **1**, 105-140 (1965)]

$$\frac{dp_z}{dt} = \frac{-4\pi n e^4 \lambda}{T} f\left(\frac{E_p}{T}\right),$$

where $f(x) = \frac{\text{erf}(\sqrt{x})}{x} - \frac{2}{\sqrt{\pi x}} e^{-x}$, and λ is the Coulomb logarithm.

We performed collisions using a small time step dt (typically 1/5000 of the expected deceleration time) and followed the momentum $p_z(t)$ of probe particles for 50 steps. After each time step, the distribution of medium particles is restored to its initial Maxwellian distribution, as assumed in the theoretical formula. The result is averaged over 62500 probe particles. Comparison between theory and simulation for different probe energies is given in the figure. The initial linear part of curve $p_z(t)$, i.e. at the high end of E_{probe} , gives us the deceleration rate of interest, that is for high initial energy electrons, demonstrating very accurate correspondence between simulation and theory. **We added a description of this test to the revised Supplementary Material (new Sec. 5).**

Reviewer #3 comment 2.4: *With the very limited provided information, the results of the simulations have to be considered as questionable. In particular, this impression is fueled by the low number of particles-per-cell (ppc) in the LCODE simulations. The authors state that they use a box radius of 9.4 mm, with a grid size of 0.19 μm , resulting in ≈ 49500 grid cells. With 5×10^4 plasma particles this results in only a single ppc per species. For example, the benchmarking tests for EPOCH or CALDER used at minimum 30 ppc up to more than 1000 ppc to achieve convergence. [Arber et al., 2015] state in their paper that for modelling collisional effects, a high number of particles per cell is critical. The number of ppc in the used LCODE simulations may increase later on due to collisional ionization. This, however, I would not expect to be statistically relevant or change the convergence of the simulations considering the large discrepancy in ppc with other well benchmarked collisional models and implementations.*

As the drawn conclusions and interpretation of the experimental data is heavily based on the simulations, it is of utmost importance to verify their convergence and reliability.

The authors list a few possible causes for the discrepancies between the measurement and the simulation. Insufficient modelling of the physics owing to a lack of particles-per-cell may actually be the most important factor. This needs to be investigated.

Our response: This is simply a misunderstanding. The initial number of plasma particles was indeed 5×10^4 . However, they were not distributed throughout the simulation window, but rather contained within the initial plasma radius $r \approx 40 \mu\text{m}$, exhibited in Fig. 3b and stated in the main text. Thus initially there were on average ~ 250 ppc for each species. The number increased subsequently in the regions and time interval of greatest interest. To correct this misunderstanding, **we added the boldfaced passages below to the description of the LCODE simulation window in Methods from which the reviewer deduced the above numbers:**

For LCODE simulations, our simulation window extended laterally out to $r = 9.4 \text{ mm}$, with grid size $0.19 \mu\text{m}$. The initial plasma consisted of 5×10^4 equal-charge macro-particles of each type **within a $40 \mu\text{m}$ radius, corresponding to an average of 250 particles per cell (ppc) per species. Because of cylindrical geometry, the number ranged from ~ 0 at the axis to ~ 500 ppc at $r = 40 \mu\text{m}$. As the plasma expanded and impact ionization proceeded, the number increased to > 1000 ppc from $r \approx 80 \mu\text{m}$ to the plasma edge throughout most of the simulated time interval. These numbers are consistent with those used for convergence tests of other PIC codes for modeling laser-plasma interactions.**^{29,30}

When Coulomb collisions came into play, the number of ppc was more than sufficient in most of the simulated volume, with the exception of a narrow region very close to the axis, whose influence on overall dynamics was negligible. The number of ppc in the LCODE simulation was well above the minimum required for convergence, and was consistent with the numbers used for convergence tests of EPOCH and CALDER. **We added citations to Perez et al., 2012 (Ref. 29) and Arber et al., 2015 (Ref. 30) in this context.**

Reviewer #3 comment 2.5: *Another peculiar finding indicating problems with the simulation results is shown in Fig. 5 c): There are a few bumps in the energy distributions, the strongest one at $\sim 800 \text{ ps}$ (see Figure, which shows a zoom into Fig. 5c). The step-like increase actually corresponds to a sudden jump in*

the distribution of the electron energy density by $\approx 18\%$. This abnormality seems nonphysical and needs to be addressed, explained and, at best, removed by the authors.

Figure 1: Snippet of Fig. 5 c)

Our response: The jumps in the energy distribution visible in figure 5(c) are indeed nonphysical. They are, unfortunately, unavoidable at reasonable computational cost. *We follow the plasma evolution for more than 3000 wakefield periods, a world record.* No other code has thus far simulated such a long evolution while resolving individual wakefield oscillations. This record comes at the expense of a trade-off between precision (determined by the number of particles) and simulation time. The most sensitive process to the number of particles per cell is wave breaking at early times (rather than particle collisions). As we have explained in the response to comment 2.4, the number of particles in the simulation grows with time, as more atoms become ionized and excited. This rapidly increases the computational load, which we address by re-scaling macro particles several times. The jumps in figure 5(c), of which the reviewer points out the largest one, occur when these sudden macro particle size and number changes are made. With the chosen number of particles per cell, the accumulated energy error suffered during wave breaking averages about 10%. This precision is comparable to that of experimental data, and we consider it acceptable. If the simulation were run with particles of the initial size, it would last more than 1 year (2d quasistatic plasma solver is not parallelizable). To speed up simulations, we double the size of macro-particles when too many of them appear due to ionization. Every other particle is discarded. Halving the number of plasma particles results in a small, instant change of integrated particle and field energies. The “jumps” become smaller if the halving operations are made later (when the number of particles is larger). In the presented run, we kept most of the “jumps” below $\sim 10\%$. This is a compromise between precision and execution time, and allows us to complete the run in 40 days. **We added the following sentences to the end of the caption of Fig. 5 to explain the origin of these jumps:**

The small jumps evident in curves (2) – (4) are nonphysical. They result from occasionally doubling macro-particle size and halving density as ionization increases particle number, in order to speed up the simulation.

Reviewer #3: 3. Minor questions regarding the experiment

Reviewer #3 comment 3.1: *The authors do not quantitatively state the temporal resolution in their measurements. They only claim their choice of propagation angle between laser and electron beam comes at a "mild cost" in temporal resolution (page 2, last line). What is this cost exactly? Does the mismatch between laser group velocity and electron beam velocity play a role?*

Our response: Several factors determine overall temporal resolution: (1) probe pulse duration; (2) geometric walk-off due to the oblique path of the probe through the plasma column; (3) optical resolution of the imaging system. See our response to Reviewer #2, comment 5 for an evaluation of the "cost" of each.

Overall temporal resolution of the presented measurements near the center of the images, where we report the column's radial profile, is ~ 15 ps, much smaller than the 100ps time steps in the data. In the revised manuscript, **we added the boldfaced passages below to state this temporal resolution quantitatively** at the end of the section "Results/ Generation of nonlinear wakes":

Nevertheless, ionization induces a large change in the vapor's refractive index η , enabling us to detect the ionization front **with ~ 15 ps time and < 40 μm space resolution (both limited primarily by imaging resolution)**, without perturbing overall energy transport dynamics significantly.

We also added the bold-faced passage below to quantify the "mild cost" of geometric walk-off alone:

By using this grazing θ ...we probed the tenuous plasma profile ~ 100 x more sensitively than with a transverse ... probe ..., at the mild cost of averaging longitudinal (z) density variations $n_e(z)$ **over $\Delta z \sim 0.5$ cm.**

Reviewer #3 comment 3.2: *I do not agree with the statement on page 3, first paragraph: "Use of an e-bunch driver eliminated strong forward-directed supercontinuum that a laser-driven nonlinear wake generates, which would saturate probe detectors in this near co-propagating pump-probe geometry." There, in principle, is the possibility to separate longitudinal probe and LWFA laser by polarization. Therefore, this statement in its generality seems to be too strong.*

Our response: Yes, there is "in principle" this possibility. Indeed a 1990s-era pump-probe experiment successfully probed *low-amplitude quasi-linear wakes* driven by tightly focused ~ 10 mJ pump pulses over *sub-mm path length*, using ~ 1 mJ probe pulses discriminated from scattered pump light via polarization alone [Siders *et al.*, *PRL* **76**, 3570 (1996)]. However, in practice, the best polarizers have rejection ratios of 10^4 . Thus all subsequent LWFA probing experiments with more energetic pump pulses have had to use frequency-shifted (e.g. doubled) probes and/or oblique probe propagation angle in addition to polarization to discriminate against scattered pump light [Downer *et al.*, *Rev. Mod. Phys.* **90**, 035002 (2018)]. In a GeV LWFA experiment with a drive laser pulse of 25 J [Gonsalves *et al.*, *PRL* **122**, 084801 (2019)] to 50 J [Wang, *NatComms* **4**, 1988 (2013)], several mJ of energy leak into a longitudinal probe line under ideal circumstances, i.e. assuming no pump depolarization from the multi-cm-long interaction of a relativistically intense pulse with a high-amplitude plasma wake, nor from optical elements. In reality, of course, that assumption is not valid. Consequently pump leakage into a longitudinal probe line is much greater than the numbers cited, in addition to being frequency-

broadened and spread over a $\sim 10^\circ$ forward cone angle. The challenges presented by depolarized, frequency-broadened, near-forward-scattered pump light even for wake probing experiments using frequency-doubled probes at oblique angles as large as 8° are well-documented in the literature [e.g. Li *et al*, *PRL* **113**, 085001 (2014)]. For probe experiments (like that presented here) in which subtle interference or diffraction features must be recorded, pump leakage must be $\ll 1\%$ of probe energy to avoid significant reconstruction errors.

In a beam driven accelerator this issue is completely absent when the plasma is created through self-ionization by the bunch. Thus we stand by the essence of the statement in question. **In deference to the reviewer's concern, we modified it slightly by substituting the bold-faced phrases below:**

Use of an e-bunch driver eliminated strong **depolarized**, forward-directed supercontinuum that a laser-driven nonlinear wake generates,²² **which is challenging to discriminate from probe light** in this near co-propagating pump-probe geometry.

REVIEWER COMMENTS

Reviewer #1 (Remarks to the Author):

The authors have answered my questions and I recommend publication after the other reviewer's questions have been answered.

Reviewer #2 (Remarks to the Author):

Dear Co-authors of manuscript NCOMMS-19-27895A:

Your revisions are thorough, you have made the manuscript clearer and you have appropriately addressed all of the issues I had raised in my report. Longer time scale investigations such as this are critical to the rep-rated cases as well as overall energy accounting over extended time intervals. Your work also inspires further meaningful associated investigations (including spectroscopic). I am recommending to the editor publication of your manuscript in its current revised form.

Review by Paul R. Bolton

Reviewer #3 (Remarks to the Author):

Zgadza et al. have improved their manuscript such that I can support publication in Nature Communications, if the last remaining question concerning the validity of their simulation approach can be answered in a satisfactory fashion (see below, under 2.2). These simulations are central to the presented conclusions and should therefore be described in all necessary detail.

Following the nicely structured reply by the authors, I will provide comments divided into the same 3 topical groups (1. General comments, 2. Validity of the simulations, and 3. Minor questions regarding the experiment.)

1 General comments

The potential impact of the here presented study largely depends on its relevance for future applications of plasma accelerators, in which the investigated energy-transport effects and dynamics become important for the outcome of the acceleration process. First and foremost, this may be the case for high repetition rate and high average power applications. In first instance, it does not matter for the energy-transport dynamics and long-term plasma evolution how the energy is deposited into a plasma wakefield (be it by a laser or beam driver). What matters is the initial spatial distribution of energy deposition and the initial condition of the ambient medium and its boundary conditions. Therefore, it is important and valid to contest and question the potential to translate the findings obtained here for the case of a beam self-ionized plasma to cases in which the initial plasma distribution is different. Such cases in which the plasma distribution is of larger diameter (compared to the transverse size of the initial wakefield) than typically generated from beam ionization are most often associated with external laser- or discharge-based ionization schemes. These schemes are of high relevance since they - in contrast to beam ionization - mitigate the detrimental and fundamental effect of beam head erosion, which leads to significantly lower total efficiency in a beam-driven plasma accelerator - a key quantity for future high-average power applications.

In the review of the initial version of the manuscript, my skepticism about the potential to directly translate the presented findings to schemes with initially larger diameter plasma columns was founded on several statements in the manuscript, all highlighted in prior communication, which indicated that the energy transport dynamics could be different in those cases.

Even though I do not agree with a number of claims made by the authors in their reply (to my comments related to this issue), I am glad to see that they rectified their ambiguous statements and significantly improved the clarity of their manuscript in this regard to my satisfaction. They now explain the relevance of their results also for larger initial plasma radii beyond those directly observed from beam ionization, enhancing the impact of the study.

Remark: Given that the chosen type of wakefield driver (laser- or beam) will not strongly change the initial distribution of energy stored in the wakefield, it is absolutely legitimate to try to understand the impact of this study on laser plasma acceleration. Since laser plasma accelerators typically feature a

much wider plasma column compared to the wakefield radius than investigated here, it is of high importance to understand the impact of a larger diameter plasma channel on the long-time dynamics. In particular in the question of long-term plasma evolution, a dogmatic separation of laser- and beam-driven plasma accelerators seems inappropriate.

2 Validity of the simulations

2.1 Limitation of the ADK model

In the revised manuscript, the authors assess the obtained field values with respect to the limitation of the ADK model and rule it out as an eventual source of errors. Although the strictly valid regime of the ADK model is exceeded, I agree with the statement that the induced errors will be only marginal because the field at the boundary of ionized and neutral lithium is below the critical field.

2.2 Applicability of the quasi-static approximation

The reply by the authors indicates a misunderstanding of my comment, I should have expressed my critique more clearly: It is not obvious that the long-term plasma evolution is correctly modelled using the quasi-static approximation. In a quasi-static code, the plasma is modelled in the longitudinal direction by a single slice only. The authors utilize this to describe the evolution at a certain longitudinal position over a long time scale. However, this approach does not allow for modelling relativistic plasma particles in the longitudinal direction and breaks down, if plasma particles are trapped in the plasma wake trailing behind the drive beam. I could imagine that e.g. phase mixing could lead to plasma particle trapping, which can not be modelled within a single slice of plasma. This was in fact my strongest concern regarding the simulations (besides the collisions, which is resolved, see the next comment). To my knowledge, this is the first time that a quasi-static code was used for such long time-scale simulations. The authors yet have to proof that their underlying assumptions which allow for a quasi-static treatment are correct. This is of utmost importance because all conclusions drawn in this work are based on these simulation results. I do realize that a benchmark against a full PIC code is hardly possible due to the immense computational costs, but it would be import to look at the longitudinal momentum distribution of the plasma particles and assess whether or not these particles become relativistic. If they do, a quasi-static treatment of the situation is invalid. This must be addressed and clarified in the manuscript.

2.3 Collisional modelling in LCODE

I agree with the authors' statement that the Takisuka-Abe model does not require to be tested, but still stand by my statement that the implementation in LCODE had to. I further agree with the statement that the previous tests

of the collisional models by Pérez et al. were performed in a different parameter regime. To my satisfaction, the authors validate their implementation by investigating the collisional deceleration of fast electrons. The authors' implementation reproduces the theory in the relevant regime well. Furthermore, the authors lift my concern regarding the number of plasma particles per cell. Indeed, the critique was based on a misunderstanding. The revised paragraph is clear and without ambiguity.

I appreciate the authors' efforts and believe it strongly increased the credibility of the simulations regarding the modelling of the collisions.

2.4 Nonphysical jumps in the energy distributions

I thank the authors for clarifying the nonphysical origin of the jumps in the energy distribution. Since the here discussed error seems to only have a quantitative and not a qualitative effect, I agree that the added statement is sufficient to resolve my concern.

3 Minor questions regarding the experiment

The answers to these questions were clear and satisfactory.

Key to type: 11 pt. *italic type: reviewer comments*

11 pt. standard type: our responses to Rev. 3 (blue)

Boldface magenta type: description of manuscript changes in response to comments

Response to Reviewer #3.

Reviewer comment: *Zgadzaj et al. have improved their manuscript such that I can support publication in Nature Communications, if the last remaining question concerning the validity of their simulation approach can be answered in a satisfactory fashion (see below, under 2.2). These simulations are central to the presented conclusions and should therefore be described in all necessary detail. Following the nicely structured reply by the authors, I will provide comments divided into the same 3 topical groups (1. General comments, 2. Validity of the simulations, and 3. Minor questions regarding the experiment.)*

Our response: We thank the reviewer for thoroughly scrutinizing our manuscript, and are pleased that we were able to address most of the concerns. Below we respond in detail only to the “last remaining question” in section 2.2, but appreciate all of the reviewer’s remarks.

Reviewer comment 2.2: Applicability of the quasi-static approximation. *The reply by the authors indicates a misunderstanding of my comment, I should have expressed my critique more clearly: It is not obvious that the long-term plasma evolution is correctly modeled using the quasi-static approximation. In a quasi-static code, the plasma is modeled in the longitudinal direction by a single slice only. The authors utilize this to describe the evolution at a certain longitudinal position over a long time scale. However, this approach does not allow for modeling relativistic plasma particles in the longitudinal direction and breaks down, if plasma particles are trapped in the plasma wake trailing behind the drive beam. I could imagine that e.g. phase mixing could lead to plasma particle trapping, which cannot be modeled within a single slice of plasma. This was in fact my strongest concern regarding the simulations (besides the collisions, which is resolved, see the next comment). To my knowledge, this is the first time that a quasi-static code was used for such long time-scale simulations. The authors yet have to proof that their underlying assumptions that allow for a quasi-static treatment are correct. This is of utmost importance because all conclusions drawn in this work are based on these simulation results. I do realize that a benchmark against a full PIC code is hardly possible due to the immense computational costs, but it would be import to look at the longitudinal momentum distribution of the plasma particles and assess whether or not these particles become relativistic. If they do, a quasi-static treatment of the situation is invalid. This must be addressed and clarified in the manuscript.*

Our response: We appreciate the clarification of this concern. Indeed, the quasi-static approximation has some applicability limits, and one of these is related to trapping plasma electrons by the plasma wave, which cannot yet be modeled by quasi-static codes. Our reasoning on this point is that such trapping, if present, should occur predominantly during the first several wakefield periods, immediately after driver passage, and in the early part of the drive bunch propagation, i.e. when the wave amplitude is highest. This space area (or time interval) is well within the reach of OSIRIS simulations, which are not limited by the quasi-static approximation (in fact, it was simulated by both codes). No particle trapping was noticed during propagation to $z = 31.9$ cm, nor up to 40 ps of wave evolution at this z . At later times, the wave amplitude is much lower, as is the electron energy, so electron trapping is impossible. The attached figure shows the velocity distribution of plasma electrons at $z = 31.9$ cm, $\Delta t = 40$ ps (the time after which we rely solely on LCODE). The longitudinal momentum p_z of forward-moving electrons with low transverse momentum ($p_r \sim < 0.5m_e c$) does not exceed $0.4m_e c = 200$ keV/c, which corresponds to kinetic energy of only 40 keV. These strongly sub-relativistic electrons cannot be trapped by the low-

amplitude wave that remains in the plasma. The few electrons at higher p_z (up to $0.8 m_{ec}$), also visible in the figure, also have large transverse momentum ($p_r > m_{ec}$), which also prevents trapping.

We plan to discuss evolution of the electron distribution function in another paper, and for this reason prefer not to include this figure in the present manuscript or its supplementary material. Instead we added the following 2 sentences to the end of the 1st paragraph of the section of the manuscript entitled “Simulations of plasma expansion”, which describes OSIRIS simulations:

No particle trapping was observed [in OSIRIS simulations] at $z \leq 31.9$ cm for up to $\Delta t = 40$ ps of wake evolution. Trapping can thus be assumed negligible at larger Δt , when wake amplitude and electron energy are lower.

There is also strong experimental evidence for absence of particle trapping under our conditions. During our runs, beam charge before and after the plasma was identical, to within sub-1% measurement accuracy, over 450 shots. Indeed, using a $v = c$ driver in sub- 10^{17} cm⁻³ plasma makes trapping essentially impossible. To summarize these points, we added the passage in boldface magenta type below to the end of sub-section “Results/Generation of nonlinear wakes” (p. 3, left column):

No witness e -bunch was injected with the driver. Moreover, non-intercepting beam charge monitors positioned before and after the plasma detected no change in beam charge (see Methods), suggesting that the wake trapped negligible charge from background plasma. Indeed, in view of the near-light speed of the driver and the sub- 10^{17} cm⁻³ plasma density, trapping of background electrons requires an external trigger, such as an optical injection pulse, under these conditions.²⁵

The new reference 25 is Deng et al., “Generation and acceleration of electron bunches from a plasma photocathode,” *Nat. Phys.* 15, 1156 (2019). We added the following new companion passage to Methods/Electron drive bunch characterization” (p.7, left column, bottom):

A pair of stripline beam position monitors, configured to measure bunch charge,⁴³ positioned before and after the plasma, measured change in bunch charge with sub-% accuracy. No change was detected.

The new reference 43 is Castorina et al., “Stripline Beam Position Monitor Modeling and Simulations for Charge Measurements, 6th In. Beam Instrum. Conf., pp. 247-250. JACOW, Geneva (2018).

REVIEWERS' COMMENTS

Reviewer #3 (Remarks to the Author):

The authors have addressed all of my concerns and answered all remaining questions with their latest revision. I recommend publication of the manuscript as is in Nature Communications and congratulate Zgadaj et al. for their excellent results.

NCOMMS-19-27895B response to Reviewer 3

Reviewer #3 (Remarks to the Author):

Reviewer #3 comment: The authors have addressed all of my concerns and answered all remaining questions with their latest revision. I recommend publication of the manuscript as is in Nature Communications and congratulate Zgadzaj et al. for their excellent results.

Our response: We thank the reviewer for valuable feedback, and are please that we were able to address all concerns.